# Complementary biosensors reveal different G-protein signaling modes triggered by GPCRs and non-receptor activators

Mikel Garcia-Marcos*

Department of Biochemistry, Boston University School of Medicine, Boston, United States

**Abstract** It has become evident that activation of heterotrimeric G-proteins by cytoplasmic proteins that are not G-protein-coupled receptors (GPCRs) plays a role in physiology and disease. Despite sharing the same biochemical guanine nucleotide exchange factor (GEF) activity as GPCRs in vitro, the mechanisms by which these cytoplasmic proteins trigger G-protein-dependent signaling in cells have not been elucidated. Heterotrimeric G-proteins can give rise to two active signaling species, Gα-GTP and dissociated Gβγ, with different downstream effectors, but how non-receptor GEFs affect the levels of these two species in cells is not known. Here, a systematic comparison of GPCRs and three unrelated non-receptor proteins with GEF activity in vitro (GIV/Girdin, AGS1/Dexras1, and Ric-8A) revealed high divergence in their contribution to generating Gα-GTP and free Gβγ in cells directly measured with live-cell biosensors. These findings demonstrate fundamental differences in how receptor and non-receptor G-protein activators promote signaling in cells despite sharing similar biochemical activities in vitro.

*For correspondence:
mgm1@bu.edu

**Competing interests:** The author declares that no competing interests exist.

## Introduction

Heterotrimeric G-proteins are ubiquitous molecular switches that transduce extracellular signals into intracellular cascades of biochemical reactions to steer cellular responses (*Gilman, 1987*). The ON/OFF state of these switches is defined by their nucleotide binding status—GDP-bound G-proteins are OFF, whereas GTP-bound G-proteins are ON. The switching between states is primarily determined by two biochemical events. The first one is nucleotide exchange of GDP for GTP, which determines the rate of activation, and the second one is hydrolysis of GTP to GDP, which determines the rate of inactivation. While these two reactions can be carried out spontaneously by G-proteins, they are tightly regulated by the enzymatic activity of other proteins in the cellular context. The G-protein regulatory mechanism controlled by G-protein-coupled receptors (GPCRs) is the best characterized to date (*Pierce et al., 2002*; *Weis and Kobilka, 2018*). GPCRs in their active conformation are guanine nucleotide exchange factors (GEFs) that catalyze nucleotide exchange on the Gα subunit of Gα-Gβγ heterotrimers. Upon GTP loading, Gβγ disengages from Gα, leading to the formation of two active signaling species, that is, Gα-GTP and free Gβγ, that can engage their respective intracellular effectors to initiate signaling cascades. In addition to GPCR-mediated activation, G-proteins are regulated by many cytoplasmic proteins with different enzymatic activities (*Sato et al., 2006*; *Siderovski and Willard, 2005*). These include proteins that accelerate the intrinsic rate of nucleotide hydrolysis by G-proteins (GTPase accelerating proteins [GAPs]) (*Ross and Wilkie, 2000*), proteins that block nucleotide exchange (guanine nucleotide dissociation inhibitors [GDIs]) (*Blumer et al., 2012*), or even GEFs that are not GPCRs (*Cismowski et al., 2000*; *DiGiacomo et al., 2018*; *Tall, 2013*; *Figure 1A*).

**Figure 1.** Approach to directly interrogate the regulation of G-protein activity by cytoplasmic proteins in cells. (**A**) Diagram of the G-protein-coupled receptor (GPCR)/G-protein activation cycle and different types of cytoplasmic G-protein regulators. (**B**) Diagram of the experimental approach used to control the activating input for G-protein regulators and simultaneously monitor the G-protein activity output. Chemically induced recruitment of various G-protein regulators to the vicinity of G-proteins at the plasma membrane is achieved through rapamycin-mediated dimerization of FKBP and FRB domains, and G-protein activity is recorded using live-cell bioluminescence resonance energy transfer (BRET) biosensors for free Gβγ or Gαi-GTP. Gβγ tagged with a split YFP (Venus) binds to the C-terminal domain of GRK3 (GRK3ct) fused to nanoluciferase (Nluc) when dissociated from Gα, resulting in BRET from Nluc to YFP. Gαi3 internally tagged with a YFP (Citrine) at the αb/αc loop binds to the synthetic sequence KB-1753 fused to Nluc when bound to GTP, resulting in BRET from Nluc to YFP.

The online version of this article includes the following source data and figure supplement(s) for figure 1:

**Figure supplement 1.** Expression of FKBP-fused G-protein regulators does not alter free Gβγ or Gαi-GTP levels under resting conditions.
**Figure supplement 1—source data 1.** Numerical data used for the graphs.

From a historical perspective, the heterotrimeric G-protein research field has relied heavily on reductionist approaches. Breaking down complex systems into defined biochemical entities that can be reconstituted and rigorously characterized in vitro has been a powerful approach to subsequently understand the mechanisms operating in cells. This approach has proven particularly successful for understanding GPCR-mediated regulation of G-proteins, a mechanism with broad biomedical implications that has been extensively characterized at the molecular level and that represents one of the most widely pursued pharmacological targets (*Sriram and Insel, 2018*). Similar biochemical reductionist approaches have also been useful to define the enzymatic activities of cytoplasmic regulators of G-proteins like GAPs, GDIs, and non-receptor GEFs, but have left several open questions about how these regulatory mechanisms operate in cells. For example, if G-protein signaling activity in cells is defined by the formation of Gα-GTP *or* free Gβγ, as each one of these species is sufficient to trigger downstream signaling, it is unclear how some GDIs and non-receptor GEFs affect G-protein signaling as a whole. For example, GDIs of the GoLoco motif family have paradoxical effects on G-proteins. On one hand, they bind to inactive, GDP-loaded Gα subunits to prevent the formation of Gα-GTP in vitro (*De Vries et al., 2000*; *Kimple et al., 2002*; *Natochin et al., 2000*). On the other hand, they also prevent the association of Gβγ with Gα-GDP (*Ghosh et al., 2003*; *Webb et al., 2005*). In fact, GoLoco motif GDIs were originally identified in a genetic screen for 'activators of G-protein signaling (AGS)' in yeast that relied on signaling readouts activated by free Gβγ (*Cismowski et al., 1999*). The situation with several non-receptor GEFs is similarly unclear. A group of non-receptor GEFs characterized by a *Gα-binding-and-activating* (GBA) motif has been shown not only to promote nucleotide exchange in vitro, but also to physically displace Gβγ from GDP-bound Gα (*Aznar et al., 2015*; *Garcia-Marcos et al., 2009*; *Maziarz et al., 2018*), raising the question of

what is the relative contribution of each mechanism, Gα-GTP formation and Gβγ release, to downstream signaling in cells. Other non-receptor GEFs like Ric-8 proteins promote nucleotide exchange on monomeric Gα but not on Gα-Gβγ heterotrimers (*Tall et al., 2003*), which has led to the contentious speculation that Ric-8 proteins might not regulate directly G-protein activity in cells, but rather work primarily as folding chaperones (*Chan et al., 2013*; *Tall et al., 2013*). Yet another non-receptor GEF, AGS1 (a.k.a. Dexras1), has been shown to activate nucleotide exchange on both monomeric Gα or Gα-Gβγ heterotrimers in vitro (*Cismowski et al., 2000*), but how it influences G-protein signaling in cells is not understood well.

Despite these gaps in mechanistic knowledge, activation of G-proteins by cytoplasmic proteins has been proven to impact various cellular processes and its dysregulation to be linked to different pathologies. This has been made particularly evident for non-receptor GEFs of the GBA family, for which the G-protein regulatory activity can be specifically disabled through mutagenesis (*Coleman et al., 2016*; *de Opakua et al., 2017*; *Garcia-Marcos et al., 2009*; *Garcia-Marcos et al., 2012*; *Maziarz et al., 2018*). This surgical approach has been leveraged to establish that G-protein regulation by GBA proteins, like GIV(a.k.a. Girdin) and DAPLE, is involved in normal physiological processes (e.g., formation of the neural tube during embryonic development), or in disease (e.g., cancer metastasis or birth defects) (*Aznar et al., 2015*; *Garcia-Marcos et al., 2015*; *Ghosh, 2015*; *Leyme et al., 2017*; *Marivin et al., 2019*). Overall, the involvement of these cytoplasmic regulators in (patho)physiological processes make imperative a more detailed understanding of their mechanisms of action in cells.

The slow progress in understanding G-protein regulation by cytoplasmic proteins compared to their regulation by GPCRs could be due to two experimental issues. One is that the G-protein regulatory function of many GPCRs can be stimulated (and inhibited) with high precision by the simple addition of extracellular ligands, whereas this is not possible for cytoplasmic regulators of G-proteins. The second experimental issue is that approaches to directly detect G-protein activity in cells have typically relied on the detection of Gα-Gβγ dissociation instead of detecting Gα-GTP. While these two signaling events correlate well in the process of GPCR-mediated G-protein activation, this is not necessarily the case for many cytoplasmic regulators of G-proteins, making evident the need for detecting both free Gβγ and Gα-GTP formation to understand how they operate in cells. Here, a cell-based approach was developed and implemented to overcome current limitations to study the mechanisms of G-protein regulation by cytoplasmic proteins. For this, the action of individual cytoplasmic regulators on their cognate G-proteins was triggered with an exogenous small molecule, and the responses evaluated with real-time biosensors for both Gα-GTP and free Gβγ. The resulting experimental system allows to precisely modulate and detect G-protein activity, similar to what can be done biochemically with purified proteins in vitro, but in the more physiologically relevant environment of the cell. This newly developed approach allowed to pinpoint key differences between the modes of G-protein signaling regulation in cells exerted by various proteins with GEF activity, including both receptor (i.e., GPCRs) and non-receptor proteins.

## Results

### Direct interrogation of G-protein activity regulation by cytoplasmic proteins in cells

To dissect the specific impact of cytoplasmic proteins on the activity of heterotrimeric G-proteins in cells, a strategy to control the signal input and simultaneously assess possible signal outputs was envisioned (*Figure 1B*). These studies were focused on Gi proteins because this is the group of heterotrimeric G-proteins for which cytoplasmic regulators have been discovered and characterized more extensively. The premise to establish control over the input is that triggering the relocalization of G-protein regulators from the cytosol to the plasma membrane would allow their action on their constitutively membrane-anchored Gi protein substrates by virtue of increasing the local concentration of the reactants. This was achieved by implementing chemically induced dimerization with rapamycin, which has been successfully applied in the past to rapidly modulate Gi signaling with some GAPs and non-receptor GEFs (*Muntean and Martemyanov, 2016*; *Parag-Sharma et al., 2016*). The simultaneous assessment of signaling outputs was carried out by using optical biosensors based on bioluminescence resonance energy transfer (BRET). Because G-protein signaling can be propagated

via either Gα-GTP or free Gβγ subunits, biosensors for each one of these two active species were implemented in parallel (*Figure 1B*). The free Gβγ biosensor is based on a previous design by *Hollins et al., 2009*; *Masuho et al., 2015*, whereas the Gαi-GTP biosensor is based on a recently described design by *Maziarz et al., 2020*. In both cases, activity is reported as an increase in BRET due to binding of fluorescent protein (FP)-tagged G-protein to a luciferase-tagged protein module that specifically binds to either dissociated Gβγ (i.e., the C-terminal region of GRK3, GRK3ct) or GTP-bound Gαi (i.e., the synthetic peptide KB-1753).

Three unrelated non-receptor GEFs were investigated: GIV (*Garcia-Marcos et al., 2009*), AGS1 (*Cismowski et al., 2000*), and Ric-8A (*Tall et al., 2003*). For comparison, the GoLoco motif of RGS12 (R12 GL), which has GDI instead of GEF activity in vitro (*Kimple et al., 2001*), was also investigated (*Figure 1B*), and the M4 muscarinic receptor (M4R), a prototypical Gi-activating GPCR, was used as an internal reference to benchmark responses. All cytoplasmic G-protein regulators were fused to the rapamycin-binding domain FKBP separated by a flexible linker. For GIV, Ric-8A and R12 GL, only the specific domains or motifs that are necessary and sufficient to regulate G-protein activity in vitro were used in the constructs (see 'Materials and methods' for details). This was done to avoid potential confounding factors for the interpretation of results, like indirect effects on G-protein signaling or undesired association with membranes in the absence of rapamycin via other domains of the proteins. Along the same lines, the prenylation CAAX motif of AGS1 was mutated to prevent its membrane targeting. To direct the FKBP-fused G-protein regulators to membranes upon rapamycin stimulation, the FRB domain that dimerizes with FKBP was fused to a membrane targeting sequence. These constructs were co-expressed in HEK293T cells along with the BRET biosensor components. It should be noted that under these experimental conditions G-proteins or their regulators are not necessarily expressed in cells at the same levels as their native counterparts, and that G-proteins and regulators are modified by fluorescent protein (FP) tagging and truncation/mutation, respectively. Despite this limitation of the approach, the effect of different regulators on G-protein activity can be precisely interrogated and directly compared under the same experimental conditions while benchmarking against GPCR-mediated responses.

FKBP-fused constructs were expressed at comparable levels (*Figure 1—figure supplement 1*). Consistent with the expectation that FKBP-fused constructs localized in the cytosol cannot effectively reach and activate G-proteins, no significant changes in BRET were observed in cells expressing the FKBP fusions in the absence of rapamycin stimulation (*Figure 1—figure supplement 1*). These constructs did not cause changes in the total levels of G-proteins expressed either. The exception was a modest increase in Gαi-GTP BRET upon expression of Ric-8A, which was paralleled by a modest increase in the protein levels of Gαi-YFP (*Maziarz et al., 2020*). The most likely explanation for the increased BRET is not the direct activation of G-proteins by Ric-8A, but an increase in non-specific donor-acceptor collisions due to the modest increase in BRET acceptor expression.

## Non-receptor GEFs display different abilities to promote Gα-GTP and/or free Gβγ formation

Rapamycin stimulation led to rapid and robust formation of free Gβγ in cells expressing either GIV or AGS1, whereas cells expressing Ric-8A did not display any response (*Figure 2*, top). The amplitude of the responses by the two non-receptor GEFs was comparable to that observed upon stimulation of the M4 muscarinic receptor (M4R), a prototypical Gi-activating GPCR, with an agonist concentration that elicits maximal activation in this assay format (*Garcia-Marcos et al., 2020*). In contrast, formation of Gαi-GTP upon rapamycin stimulation was only detected in cells expressing AGS1 but not in cells expressing GIV or Ric-8A (*Figure 2*, top). The Gαi-GTP response with AGS1 was comparable to that observed upon M4R stimulation. Consistent with a previous report (*Maziarz et al., 2020*), R12 GL, which has GDI activity in vitro, also led to an increase in free Gβγ but caused no change in Gαi-GTP levels (*Figure 2*). The increase in free Gβγ was somewhat smaller than that observed with GIV or AGS1. To further characterize and compare the mechanism of G-protein activation by non-receptor GEFs and GPCRs, we investigated the effect of pertussis toxin (PTX) on the Gβγ responses observed upon stimulation of GIV, AGS1, or M4R. PTX ADP-ribosylates a cysteine residue in the C-terminal tail of Gαi proteins, which precludes their binding to and activation by GPCRs. As expected, PTX completely suppressed the Gβγ response upon M4R stimulation (*Figure 2—figure supplement 1*). As for the two non-receptor GEFs, PTX did not affect the Gβγ response elicited by GIV, but greatly diminished the response by AGS1 (*Figure 2—figure*

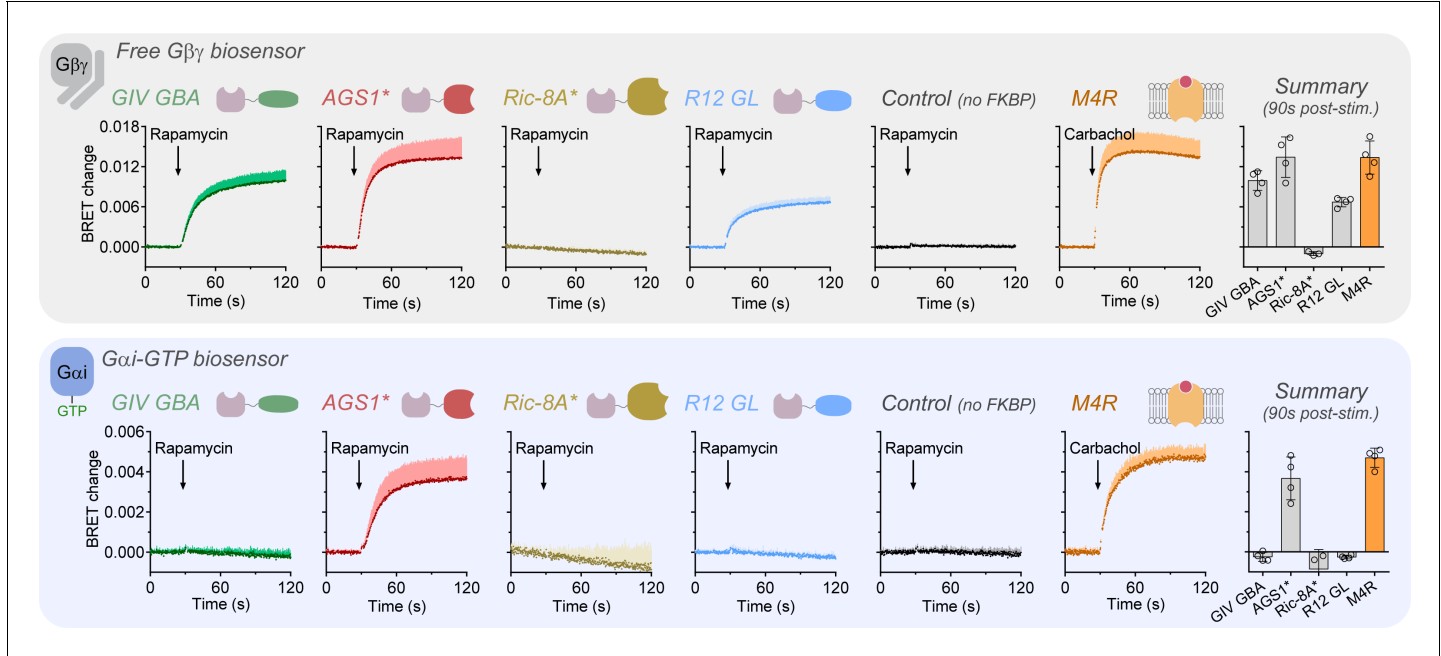

**Figure 2.** Non-receptor guanine nucleotide exchange factors (GEFs) display different abilities to promote Gα-GTP and/or free Gβγ formation. HEK293T cells expressing the components of the bioluminescence resonance energy transfer (BRET) biosensor for free Gβγ (top) or Gαi-GTP (bottom), the membrane-anchored FRB construct, and the indicated FKBP-fused G-protein regulators GIV GBA, AGS1*, Ric-8A*, or R12 GL were stimulated with rapamycin (0.5 µM) at the indicated time during kinetic BRET measurements. Stimulation of ectopically expressed M4 muscarinic receptor (M4R) with carbachol (100 µM) was done as a reference condition, and rapamycin stimulation of cells not expressing FKBP-fused constructs was done as a negative control. Bar graphs on the right summarize the BRET changes 90 s after addition of rapamycin or carbachol. Mean ± SD, n = 3–4. In the kinetic traces, the SD is displayed as bars of lighter color tone than data points and only in the positive direction for clarity.

The online version of this article includes the following source data and figure supplement(s) for figure 2:

**Source data 1.** Numerical data used for the upper panel (free Gβγ biosensor).
**Source data 2.** Numerical data used for the lower panel (Gαi-GTP biosensor).
**Figure supplement 1.** G-protein activation by a GPCR or by AGS1*, but not by GIV GBA, is inhibited by pertussis toxin (PTX).
**Figure supplement 1—source data 1.** Numerical data used for the graphs.
**Figure supplement 2.** GIV-CT has the same G-protein activating properties as GIV GBA motif.
**Figure supplement 2—source data 1.** Numerical data used for the upper panel (free Gβγ biosensor).
**Figure supplement 2—source data 2.** Numerical data used for the lower panel (Gαi-GTP biosensor).

*supplement 1*). The lack of effect of PTX on the GIV-mediated response is consistent with the lack of involvement of the C-terminus of Gαi in binding to GIV (*de Opakua et al., 2017*).

Together, these results highlight marked differences in G-protein activation mechanisms among non-receptor GEFs. AGS1 mimics GPCRs in that it activates proportionately Gα-GTP and free Gβγ formation, and that its action is suppressed by PTX. In contrast, GIV and Ric-8A fail to promote detectable Gα-GTP formation, despite possessing GEF activity in vitro. For Ric-8A, this could be explained by previous observations that it cannot promote nucleotide exchange on G-protein heterotrimers (*Tall et al., 2003*), which might be the predominant G-protein species in cells (*Krumins and Gilman, 2006*). The lack of Gβγ formation by Ric-8A would also be in agreement with this interpretation. The lack of measurable Gαi-GTP formation by GIV is more puzzling for several reasons. First, the release of Gβγ under the same experimental conditions demonstrates that GIV can rapidly act on Gαi within G-protein heterotrimers in cells. Second, previous work has shown that GIV can promote the formation of Gαi-GTP in cells by using antibodies that specifically recognize this species (*Lin et al., 2014*; *Lopez-Sanchez et al., 2014*; *Midde et al., 2015*). And third, even with a very similar experimental paradigm of chemically induced membrane recruitment, GIV has been shown to inhibit cAMP (*Maziarz et al., 2018*), presumably via inhibition of adenylyl cyclase by Gαi-GTP. The next sections focus on addressing this puzzle on the mechanism of G-protein activation by GIV.

## GIV-CT has the same G-protein activating properties as GIV GBA motif

One potential caveat of the experiments with GIV above is that the construct used contained only its GBA motif. Although it is unlikely that this would explain the lack of Gαi-GTP formation, because the GBA motif is sufficient to promote nucleotide exchange in vitro (*de Opakua et al., 2017*), this issue was investigated more thoroughly by using a larger GIV construct. Experiments in this assay format with full-length GIV are not feasible because the protein is >250 KDa and contains multiple domains that associate with membranes or cytoskeletal components. Instead, a C-terminal region of 210 amino acids (1660–1870, GIV-CT) was used. GIV-CT not only fully recapitulates the properties of GIV GBA motif in activating G-proteins in vitro (*de Opakua et al., 2017*), but also recapitulates the properties of full-length GIV in promoting G-protein-dependent signaling in cells (*Ma et al., 2015a*; *Midde et al., 2015*). Rapamycin-induced recruitment of GIV-CT induced an increase in free Gβγ levels similar to that caused by GIV GBA (*Figure 2—figure supplement 2*). Also like GIV GBA, it failed to elicit any detectable increase in Gαi-GTP, suggesting that the lack of Gαi-GTP response is not an inherent defect of the shorter construct.

## cAMP dampening by GIV's GBA motif is blocked upon Gβγ scavenging

Suppression of cAMP through direct inhibition of adenylyl cyclase activity is what originally defined the Gi subfamily of G-proteins. Although this is widely attributed to the action of Gαi-GTP on adenylyl cyclases, Gβγ can also directly modulate the production of cAMP by adenylyl cyclases (*Sadana and Dessauer, 2009*; *Sunahara et al., 1996*). GIV has been previously shown to decrease cAMP levels in cells (*Maziarz et al., 2018*; *Midde et al., 2015*), and the results presented above (*Figure 2*) indicate that it promotes the formation of detectable levels of Gβγ but not of Gαi-GTP. Together, the above prompted the investigation of whether GIV-induced cAMP dampening is mediated though Gβγ. For this, the effect of GIV recruitment to membranes on forskolin-induced cAMP was determined with or without co-expression of the C-terminal region of GRK2 (GRK2ct) (*Figure 3*). GRK2ct binds with high affinity to free Gβγ subunits and precludes their binding to effectors (*Koch et al., 1994*). Consistent with previous findings by *Maziarz et al., 2018*, GIV recruitment to membranes led to a decrease in cAMP (*Figure 3*). This effect of GIV was efficiently suppressed by expression of GRK2ct (*Figure 3*), suggesting that it is primarily mediated by the formation of free Gβγ subunits rather than by Gαi-GTP.

## Recruitment of GIV's GBA motif to RTKs promotes Gβγ release but not Gαi-GTP formation

*Kalogriopoulos et al., 2020* have recently proposed that GIV facilitates EGFR-mediated phosphorylation of Gαi by binding simultaneously to the inactive G-protein and the active, auto-phosphorylated RTK. Formation of this complex leads to Gαi phosphorylation by EGFR, which in turn promotes GTP loading by accelerating nucleotide exchange (*Figure 3—figure supplement 1*, top). This posits that a mechanism by which GIV promotes Gαi-GTP formation in cells is through GBA motif-dependent recruitment of Gαi to the vicinity of EGFR. To directly test this model (*Figure 3—figure supplement 1*, top), Gαi-GTP BRET was determined upon EGF stimulation in HEK293T cells expressing EGFR and GIV. In initial experiments with cells expressing GIV-CT, which contains both the G-protein and the RTK binding regions (*Lin et al., 2014*), no Gαi-GTP or free Gβγ formation was detected upon EGF stimulation (*data not shown*), suggesting that recruitment of GIV to EGFR is inefficient under these conditions. As an alternative to overcome this limitation, we addressed the impact of GIV-dependent recruitment of Gαi to EGFR by fusing GIV's GBA motif to the adaptor protein Grb2, which is efficiently recruited to active EGFR (*Lowenstein et al., 1992*; *Figure 3—figure supplement 1*). In cells expressing the Grb2-GBA fusion, EGF stimulation led to an increase of free Gβγ but not of Gαi-GTP (*Figure 3—figure supplement 1*). The Gβγ response was not recapitulated when a Grb2-GBA construct bearing the Gαi binding-deficient mutation F1685A (FA) of GIV (*de Opakua et al., 2017*; *Garcia-Marcos et al., 2009*) was used (*Figure 3—figure supplement 1*), indicating that the observed response is specifically caused by GIV's GBA motif and not by other EGFR triggered signaling events. These results show that, while an active EGFR-GIV complex engages Gi proteins to promote the release of Gβγ, it is still inefficient in promoting the formation of Gαi-GTP.

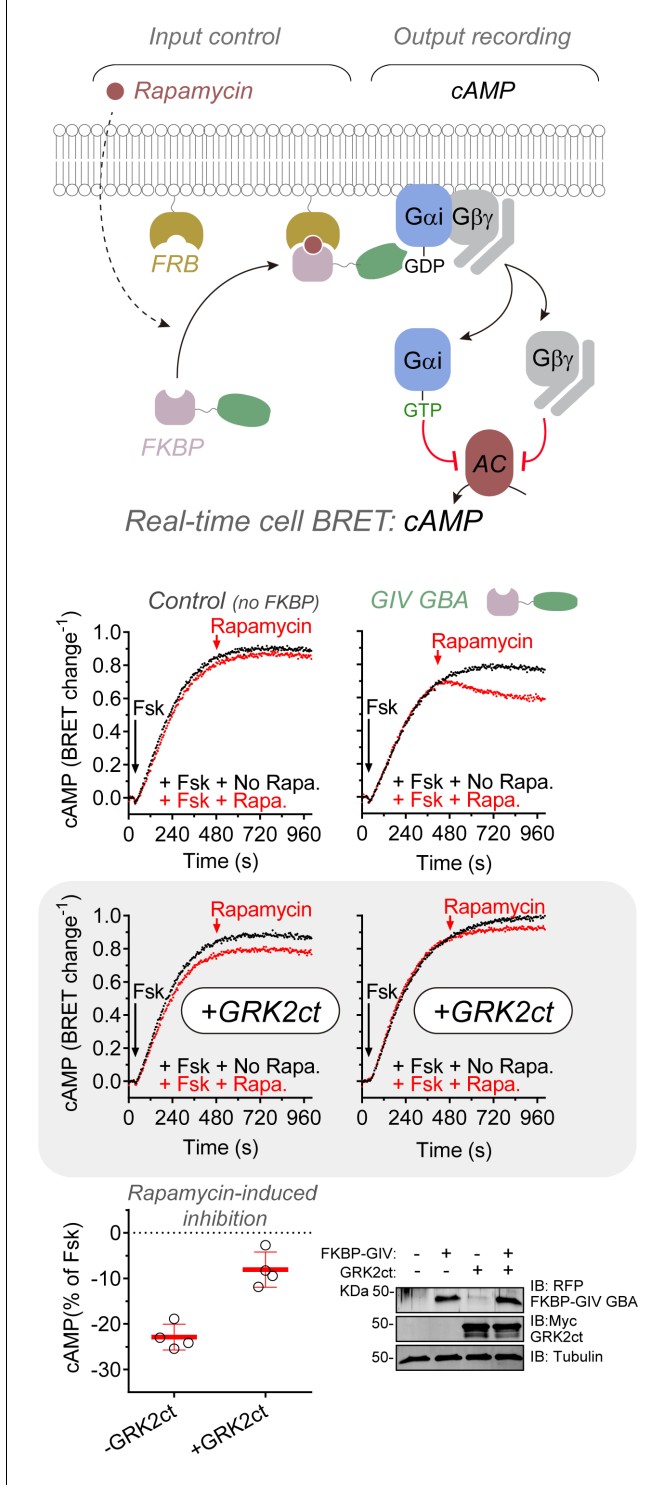

**Figure 3.** GIV-mediated cAMP dampening is prevented upon Gβγ scavenging. HEK293T cells were transfected with plasmids for the expression of the cAMP sensor Nluc-EPAC-VV, the membrane-anchored FRB construct, and FKBP-fused GIV GBA or an empty plasmid in the presence or absence of GRK2ct as indicated. Cells were stimulated with forskolin (black) or sequentially with forskolin and rapamycin (red) at the indicated times during kinetic BRET measurements. Forskolin (Fsk) = 3 μM; rapamycin = 0.5 μM. Time traces are from one representative experiment, and the quantification of rapamycin-induced inhibition of the forskolin cAMP response shown presented in the scatter plot on the bottom left is the mean ± SD of four independent experiments. A representative immunoblot confirming the expression of GIV GBA and GRK2ct is shown on the bottom right.

*Figure 3 continued on next page*

*Figure 3 continued*

The online version of this article includes the following source data and figure supplement(s) for figure 3:

**Source data 1.** Numerical data used for the lower panel (Gαi-GTP biosensor).
**Figure supplement 1.** Recruitment of GIVGBA motif to RTKs promotes Gβγ release but not Gαi-GTP formation.
**Figure supplement 1—source data 1.** Numerical data used for the upper graphs (free Gβγ biosensor).
**Figure supplement 1—source data 2.** Numerical data used for the lower graphs (Gαi-GTP biosensor).

## Gβγ release by GIV is insensitive to cellular GTP depletion

It is possible that (i) GIV promotes the formation of Gαi-GTP below the detection levels of the BRET assay used above, and that (ii) this in turn contributes to the formation of larger levels of free Gβγ. The former is suggested by previous evidence of GIV-dependent Gαi-GTP formation in cells by using an antibody-based approach (*Maziarz et al., 2018*; *Midde et al., 2015*), which might be more sensitive than the BRET assay above. The latter is suggested by the observation that R12 GL promotes a smaller increase of free Gβγ than GIV (*Figure 2*), even though R12 GL has an affinity for Gαi about 10 times higher than that of GIV (*de Opakua et al., 2017*). It is therefore conceivable that GIV utilizes mechanisms other than just physical displacement by mass action to achieve formation of free Gβγ more efficiently than R12 GL. For example, after displacing Gβγ from Gαi, GIV might weakly promote Gαi-GTP formation to sustain the dissociated status of Gβγ, which R12 GL could not because it is a GDI. This point was addressed by investigating the requirement of GTP for GIV-mediated Gβγ responses using a nucleotide depletion protocol in semi-permeabilized cells previously described by *Qin et al., 2011*; *Qin et al., 2008*. BRET responses triggered by different regulators were compared in cells depleted of nucleotides or replenished with near-physiological levels of GTP (0.25 mM). GIV-induced Gβγ responses were very similar in the presence or absence of added GTP after the nucleotide depletion protocol (*Figure 4*). Similar observations were made for R12 GL-induced Gβγ BRET responses, which are not expected to require GTP. In contrast, agonist-stimulated GPCR Gβγ BRET responses were larger in the GTP-replenished condition (*Figure 4*). Although it is unclear if the GPCR response observed in the absence of GTP addition is due to incomplete nucleotide depletion and/or a G-protein rearrangement that occurs upon engagement with active GPCRs in the absence of nucleotides (*Chung et al., 2011*; *Rasmussen et al., 2011*), these results indicate that Gβγ release induced by GIV is largely independent of the presence of physiological levels of GTP.

## GIV GBA does not suppress GPCR-mediated Gα-GTP formation like R12 GL

Data presented above shows that GIV GBA and R12 GL display similar G-protein signaling properties in cells despite having striking different behavior in vitro—that is, the former is a GEF and the latter is a GDI. Next, the effects of GIV GBA and R12 GL on Gα-GTP formation were compared under conditions of GPCR stimulation, which represent a regime of high nucleotide exchange not present in the previously investigated resting conditions. After stimulation of M4R with carbachol, addition of rapamycin led to a rapid drop of Gαi-GTP in cells expressing R12 GL compared to controls, whereas no significant change in Gαi-GTP was observed in cells expressing GIV GBA (*Figure 5*). These results suggest that the GDI activity of R12 GL efficiently sequesters monomeric Gα-GDP, reducing the pool of Gα(GDP)-Gβγ trimeric substrate used by the GPCR to sustain the higher steady-state levels of Gα-GTP achieved upon stimulation. In contrast, GIV GBA, which is even more efficient than R12 GL in dissociating Gα-Gβγ complexes (*Figure 2*), does not seem to significantly prevent the utilization of Gα(GDP)-Gβγ by GPCRs, suggesting that it does not lead to sequestration of monomeric Gα-GDP. As opposed to what occurs with a GDI like R12 GL, which locks Gα in its GDP-bound state, GIV-bound Gα can exchange nucleotides. Then, GIV would dissociate from the G-protein upon transient formation of Gαi-GTP, which would be rapidly converted into Gα-GDP to replenish Gα(GDP)-Gβγ levels.

## GIV GBA does not hinder GPCR-mediated activation of G-proteins

The sequestration model proposed above was further investigated by characterizing how different cytoplasmic regulators affect GPCR-mediated G-protein activation. Essentially, three qualitative

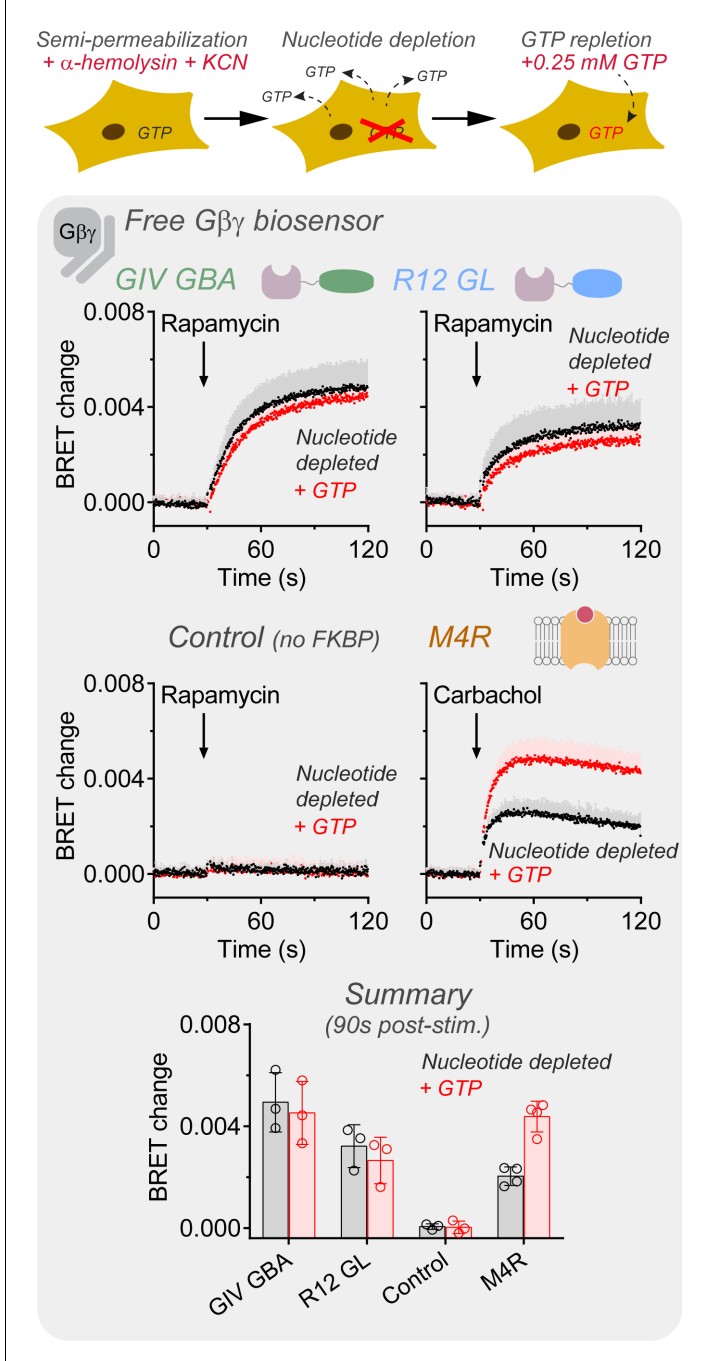

**Figure 4.** Gβγ release by GIV is insensitive to cellular GTP depletion. HEK293T cells expressing the components of the BRET biosensor for free Gβγ, the membrane-anchored FRB construct, and FKBP-fused GIV GBA or R12 GL were depleted of nucleotides (black) or replenished with GTP (0.25 mM, red) as indicated in 'Materials and methods'. Cells were stimulated with rapamycin (0.5 µM) at the indicated time during kinetic BRET measurements. Stimulation of ectopically expressed M4R with carbachol (100 µM) was done as a reference condition, and rapamycin stimulation of cells not expressing FKBP-fused constructs was done as a negative control. The bar graph on the bottom summarizes the BRET changes 90 s after addition of rapamycin or carbachol. Mean ± SD, n = 3–4. In the kinetic traces, the SD is displayed as bars of lighter color tone than data points and only in the positive direction for clarity.

The online version of this article includes the following source data for figure 4:

**Source data 1.** Numerical data used for the graphs.

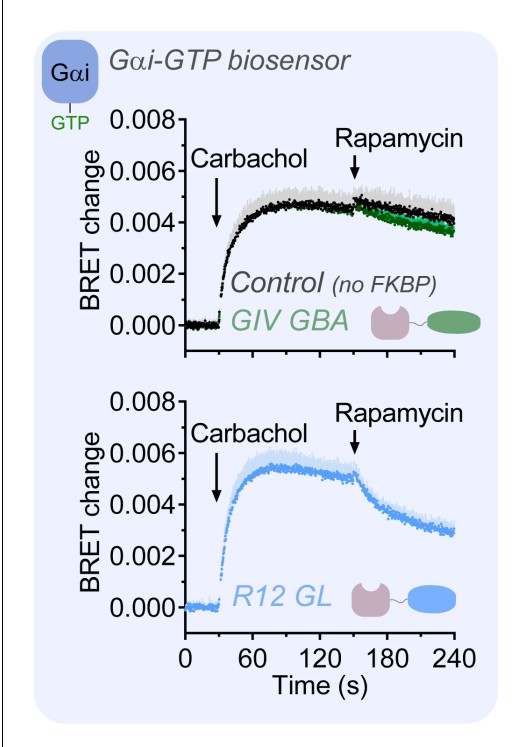

**Figure 5.** R12 GL, but not GIV GBA, suppresses GPCR-mediated Gα-GTP formation. HEK293T cells expressing the components of the BRET biosensor for Gαi-GTP, the membrane-anchored FRB construct, M4R, and FKBP-fused GIV GBA or R12 GL were sequentially stimulated with carbachol (100 µM) and rapamycin (0.5 µM) at the indicated times during kinetic BRET measurements. Mean ± SD, n = 4. SD is displayed as bars of lighter color tone than data points and only in the positive direction for clarity.

The online version of this article includes the following source data for figure 5:

**Source data 1.** Numerical data used for the graphs.

scenarios were proposed and tested considering that the availability of Gα(GDP)-Gβγ complexes is limiting their utilization by GPCRs to generate Gαi-GTP and free Gβγ (*Figure 6A*). In each scenario, the action of cytoplasmic regulators is triggered before GPCR stimulation, which leads to a new dynamic equilibrium between Gα(GDP)-Gβγ and active Gα/Gβγ species prior to receptor-mediated activation. In scenario one, the GDI R12 GL reduces the availability of Gα(GDP)-Gβγ by dissociating Gα from Gβγ and locking it in the Gα-GDP state. In two, GIV GBA does not reduce the availability of Gα(GDP)-Gβγ because the association with Gα-GDP can be reversed upon nucleotide exchange. In three, the GEF AGS1 reduces the availability of Gα(GDP)-Gβγ by competing for it with GPCRs to independently generate Gα-GTP and free Gβγ through multiple turnover cycles. Consistent with these proposed models, the formation of Gαi-GTP and free Gβγ upon M4R stimulation with carbachol were reduced by pretreatment with rapamycin in cells expressing R12 GL or AGS1 but not in cells expressing GIV (*Figure 6B,C*). The total Gβγ BRET change observed after GIV recruitment and GPCR stimulation (time 240 s, *Figure 6C*) was larger than in any of the other conditions, indicating that the reduced GPCR response by R12 GL or AGS1 was not due to reaching a maximum signal. Furthermore, if the association of GIV GBA with Gα-GDP can be reverted as proposed, one would also expect that its ability to generate free Gβγ would be hampered after GPCR stimulation due to competitive removal of available Gα (GDP)-Gβγ. In other words, once a GPCR is activated, formation of Gα(GDP)-GIV (and subsequent generation of free Gβγ) would be disfavored because the equilibrium of G-protein complexes is shifted toward formation of receptor-Gα(GDP)-Gβγ. Indeed, the Gβγ response to GIV GBA was diminished in cells in which M4R had been pre-stimulated with carbachol (*Figure 6—figure supplement 1*). Taken together, these results suggest that GIV, despite efficiently triggering G-protein signaling via Gβγ, does oppose activation of G-proteins by GPCRs.

## Discussion

This work introduces an experimental paradigm to investigate the regulation of G-proteins in cells with high precision and for a wide range of regulators beyond GPCRs. The experimental framework presented here is poised to complement other approaches that have traditionally been used to study G-protein signaling, like reductionist biochemical assays with purified proteins and genetic manipulations for more complex systems like cultured cells or whole organisms. Experiments with purified proteins allow gaining detailed insights into molecular mechanisms but can suffer from lack of physiological context, whereas genetic manipulations can reveal functionality in more physiological contexts but can fall short in delivering mechanistic detail at the molecular level. The approach presented here leverages some advantages of reductionism, in that it permits studying isolated G-protein biochemical events like nucleotide binding status and subunit dissociation upon

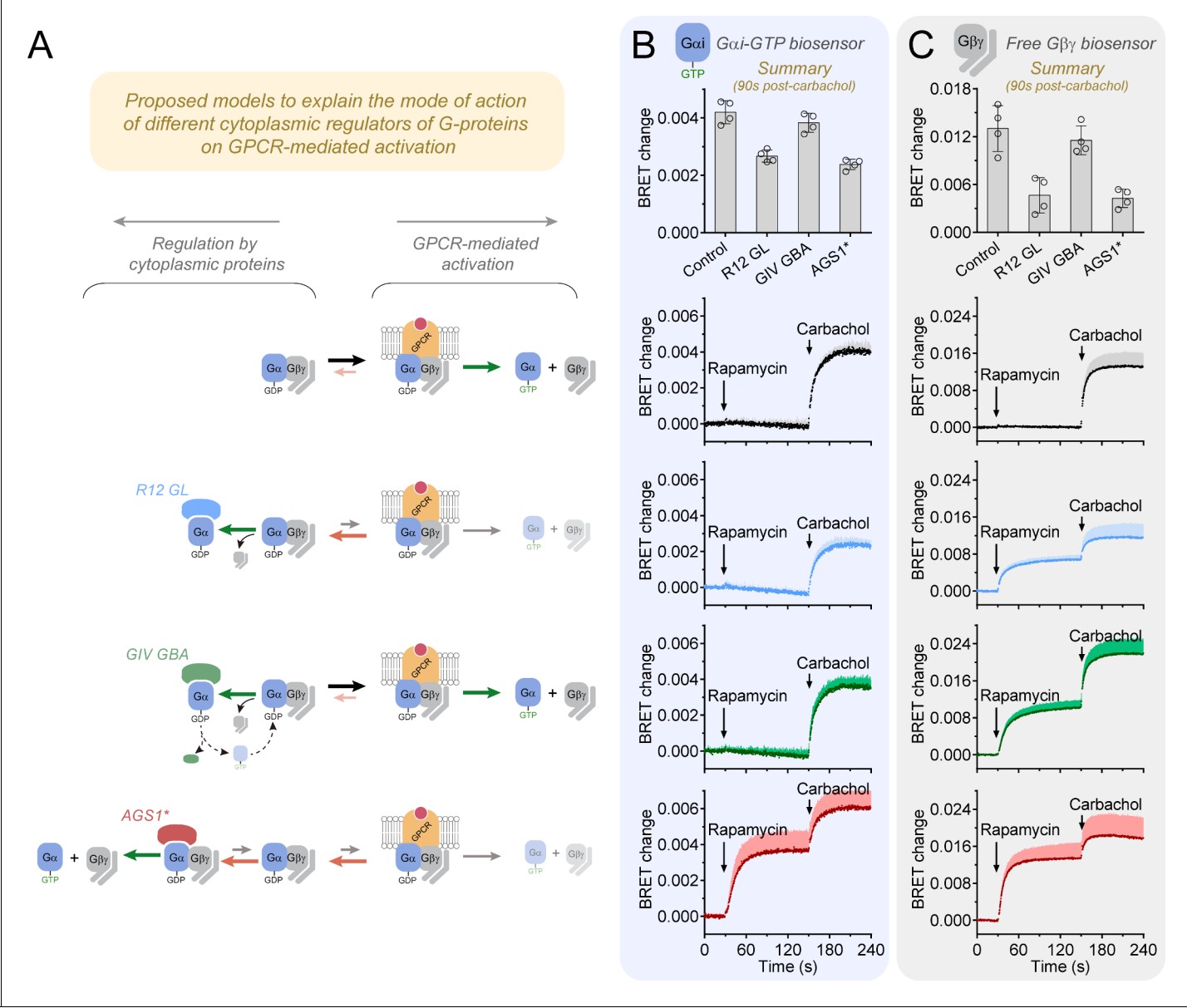

**Figure 6.** R12 GL and AGS1*, but not GIV GBA, hinder the activation of G-proteins by a GPCR. (**A**) Proposed models to explain the mode of action of different cytoplasmic regulators of G-proteins on GPCR-mediated activation. (**B, C**) HEK293T cells expressing the components of the BRET biosensor for Gαi-GTP (**B**) or free Gβγ (**C**), M4R, the membrane-anchored FRB construct, and the indicated FKBP-fused G-protein regulators (R12 GL, GIV GBA, or AGS1*) were sequentially stimulated with rapamycin (0.5 μM) and carbachol at the indicated times during kinetic BRET measurements. Stimulation of cells not expressing FKBP-fused constructs with rapamycin was done as a control. Bar graphs on the top summarize the BRET changes 90 s after addition of carbachol. Mean ± SD, n = 4. In the kinetic traces, the SD is displayed as bars of lighter color tone than data points and only in the positive direction for clarity.

The online version of this article includes the following source data and figure supplement(s) for figure 6:

**Source data 1.** Numerical data used for the graphs on the left (Gαi-GTP biosensor).

**Source data 2.** Numerical data used for the graphs on the right (free Gβγ biosensor).

**Figure supplement 1.** Pre-stimulation of a GPCR diminishes GIV-induced Gβγ release.

**Figure supplement 1—source data 1.** Numerical data used for the graph.

modulation by specific protein regulators, but within the more physiological context of the cell. In this sense, it could be considered a 'cell-based reductionist' approach. The benefits of this are demonstrated here by the discovery that G-protein signaling modes in cells can differ greatly among

proteins with similar G-protein regulatory functions in vitro, including differences between receptor and non-receptor GEFs or even among different non-receptor GEFs (*see further discussion below*). Also, the interplay between GPCRs and different cytoplasmic regulators in controlling G-protein activity in cells was further clarified. Like with biochemical reductionist approaches, a limitation of the studies presented here and the approach in general if considered alone is that they still rely on non-native conditions, like the use of overexpression, protein fragment fusions, or artificial means to alter the subcellular localization of proteins. It is in the context of complementarity with other well-established approaches like in vitro biochemistry and genetics that this limitation is outweighed by the additional mechanistic insights it can provide.

The in-depth characterization of the mechanisms by which non-GPCR proteins of the GBA family trigger G-protein signaling in cells exemplifies well the additional insights that can be gained through the approach presented in this work. On one hand, it was known from in vitro biochemical experiments that GBA motifs have GEF activity on Gαi proteins and that they also promote the dissociation of Gβγ from Gαi-GDP (*Aznar et al., 2015*; *Coleman et al., 2016*; *de Opakua et al., 2017*; *Garcia-Marcos et al., 2011a*; *Garcia-Marcos et al., 2010*; *Garcia-Marcos et al., 2009*; *Garcia-Marcos et al., 2012*; *Garcia-Marcos et al., 2011c*; *Marivin et al., 2019*; *Maziarz et al., 2018*). On the other hand, genetic approaches had also established that the GBA motif of some proteins, like GIV and DAPLE, controls G-protein signaling in cells or whole organisms (*Aznar et al., 2015*; *Garcia-Marcos et al., 2011a*; *Garcia-Marcos et al., 2010*; *Garcia-Marcos et al., 2009*; *Garcia-Marcos et al., 2012*; *Landin Malt et al., 2020*; *Leyme et al., 2016*; *Leyme et al., 2017*; *Leyme et al., 2015*; *Lin et al., 2014*; *Lo et al., 2015*; *Lopez-Sanchez et al., 2014*; *Ma et al., 2015b*; *Marivin et al., 2019*; *Midde et al., 2015*; *Sasaki et al., 2015*). Thus, an unresolved question so far had been the relative contribution of the two potentially overlapping mechanism of G-protein activation-mediated GBA motifs, that is, generation of Gαi-GTP and formation of free Gβγ. A key conclusion of the studies presented here (*Figure 2*, *Figure 4*) is that GIV, and most likely other GBA proteins, activates G-protein signaling in cells primarily through the formation of free Gβγ rather than through the formation of Gαi-GTP, despite its GEF activity in vitro. GIV enhances nucleotide exchange in vitro approximately 2.5- to 3-fold (*Garcia-Marcos et al., 2011a*; *Garcia-Marcos et al., 2010*; *Garcia-Marcos et al., 2009*; *Garcia-Marcos et al., 2012*), which is very similar to the approximately 3-fold enhancement mediated by AGS1 under similar conditions in vitro (*Cismowski et al., 2000*). In contrast, the studies described here reveal that AGS1 efficiently triggers the formation of Gαi-GTP in cells, whereas GIV does not (*Figure 2*). Similarly, the enhancement of nucleotide exchange on Gi by a GPCR (e.g. α2 adrenergic receptor) in vitro is 3- to 6-fold (*Cerione et al., 1986*; *Kurose et al., 1991*), which is stronger than that of GIV or AGS1 but still within the same order of magnitude. Thus, the results presented here indicate that the relative GEF activity in vitro across different G-protein activators does not correlate well with their ability to generate Gα-GTP in cells. These observations not only provide mechanistic insights into G-protein activation by GBA proteins that were not evident from experiments using other approaches, but also highlight the importance of elucidating the molecular mechanisms of G-protein regulation in a cellular context.

The conclusion that GIV promotes G-protein signaling in cells primarily through the formation of free Gβγ prompts the re-evaluation of previous signaling studies in cells. In fact, the vast majority of signaling readouts regulated by the GBA motif of GIV (or DAPLE) have been shown or are known to be controlled by Gβγ rather than by Gαi-GTP. This includes activation of the PI3K-Akt axis (*Aznar et al., 2015*; *Bhandari et al., 2015*; *Garcia-Marcos et al., 2010*; *Garcia-Marcos et al., 2009*; *Garcia-Marcos et al., 2012*; *Gupta et al., 2016*; *Leyme et al., 2016*; *Leyme et al., 2017*; *Leyme et al., 2015*; *Ma et al., 2015a*), p114RhoGEF-mediated activation of RhoA (*Marivin et al., 2019*), or Rac1 activation (*Aznar et al., 2015*). Although GBA-mediated inhibition of adenylyl cyclase (*Aznar et al., 2015*; *Maziarz et al., 2018*; *Midde et al., 2015*) could have been explained as an effect of Gαi-GTP, evidence presented here (*Figure 3*) strongly suggests that this is indeed mediated by Gβγ as well. Thus, although previous evidence has identified detectable levels of Gαi-GTP in cells upon GIV action (*Lin et al., 2014*; *Lopez-Sanchez et al., 2014*; *Midde et al., 2015*), these are probably much lower than those achieved upon activation of a GPCR and insufficient to drive robust signaling directly. Thus, the functional role of the weak GEF activity of proteins with a GBA motif remains to be elucidated, whereas efficient release of Gβγ subunits seems to be main mechanism by which this type of G-protein activator promotes signaling in cells.

Results shown here (*Figure 5*, *Figure 6*) also indicate that proteins with a GBA motif differ mechanistically from other regulators that promote the dissociation of Gβγ from G-protein heterotrimers, like GDIs that contain a GoLoco motif. This in turn might be important for the interplay between GBA proteins and GPCRs in regulating G-proteins. The data suggest that activation of G-proteins by GIV, which in many cases is triggered by surface receptors different from GPCRs (*Garcia-Marcos et al., 2015*; *Leyme et al., 2015*; *Lopez-Sanchez et al., 2014*), might operate without opposing efficient GPCR-mediated G-protein activation. In contrast, GDIs with a GoLoco motif efficiently suppress GPCR-mediated activation of G-proteins. Interestingly, the artificial system implemented here to recruit GIV to membranes mimics the recruitment of native GIV from the cytosol to membranes when it binds non-GPCR surface receptors upon ligand stimulation (*Ghosh et al., 2010*; *Leyme et al., 2015*), which might be the mechanism by which GIV action on G-proteins is controlled under native conditions (*Parag-Sharma et al., 2016*). Nevertheless, the role of GIV in the interplay between GPCR and non-GPCR receptors in G-protein regulation needs to be characterized in more detail in the future.

Beyond insights gained in the understanding of GBA-mediated mechanisms of G-protein signaling regulation, the present study also provides other useful information. For example, it was shown that pertussis toxin is not a generic inhibitor of all mechanisms of Gi activation or a specific inhibitor of GPCR-mediated activation of Gi proteins (*Figure 2—figure supplement 1*), which warranties caution in the interpretation of past and future experiments with this widely used reagent. In addition, the chemogenetic tools presented here could be easily adapted for other applications, like synthetic biology approaches to turn different modes of G-protein signaling ON and OFF.

In summary, the combination of chemogenetics and optical biosensors presented here has demonstrated the potential to become an experimental paradigm to expand how we study and understand signal transduction mechanisms mediated by heterotrimeric G-proteins.

# Materials and methods

**Key resources table**

| Reagent type (species) or resource | Designation | Source or reference | Identifiers | Additional information |
|---|---|---|---|---|
| Cell line (*Homo sapiens*) | HEK293T cells | ATCC | CRL3216 | |
| Antibody | α-Tubulin (mouse monoclonal) | Sigma | T6074 | Immunoblotting dilution (1: 2500) |
| Antibody | RFP (rabbit polyclonal) | Rockland | 600-401-379 | Immunoblotting dilution (1: 1000) |
| Antibody | GFP (mouse monoclonal) | Clontech/Takara Bio | Cat# 632380 | Immunoblotting dilution (1: 1000) |
| Antibody | Hemagglutinin (HA) tag (clone 12CA5) (mouse monoclonal) | Roche | Cat# 11583816001 | Immunoblotting dilution (1: 1000) |
| Antibody | MYC tag (9B11) (mouse monoclonal) | Cell Signaling | Cat# 2276 | Immunoblotting dilution (1: 1000) |
| Antibody | Gαi3 (rabbit polyclonal) | Santa Cruz Biotechnology | Cat# sc-262 | Immunoblotting dilution (1:250) |
| Antibody | Pan-Gβ (rabbit polyclonal) | Santa Cruz Biotechnology | Cat# sc-261 | Immunoblotting dilution (1: 250) |
| Antibody | Goat anti-rabbit Alexa Fluor 680 (goat polyclonal) | Life Technologies | Cat# A21077 | Immunoblotting dilution (1:10,000) |

*Continued on next page*

Continued

| Reagent type (species) or resource | Designation | Source or reference | Identifiers | Additional information |
|---|---|---|---|---|
| Antibody | Goat anti-mouse IRDye 800 (goat polyclonal) | LiCor | Cat# 926–32210 | Immunoblotting dilution (1:10,000) |
| Recombinant DNA reagent | pmRFP-FKBP-pseudojanin (plasmid) | Addgene | Cat# 37999 | |
| Recombinant DNA reagent | pmRFP-FKBP-GIV GBA (plasmid) | *Parag-Sharma et al., 2016* | | Contains human GIV aa1660-1705 |
| Recombinant DNA reagent | pmRFP-FKBP-AGS1* (plasmid) | This paper | | Contains rat AGS1 with C278S mutation. See details in 'Plasmids' section of 'Materials and methods' |
| Recombinant DNA reagent | pmRFP-FKBP-Ric-8A* (plasmid) | This paper | | Contains rat Ric-8A aa12-492 See details in 'Plasmids' section of 'Materials and methods' |
| Recombinant DNA reagent | pmRFP-FKBP-R12 GL (plasmid) | *Maziarz et al., 2020* | | Contains mouse RGS12 aa1185-1221 See details in 'Plasmids' section of 'Materials and methods' |
| Recombinant DNA reagent | Lyn11-FRB (plasmid) | *Parag-Sharma et al., 2016* | | |
| Recombinant DNA reagent | pcDNA3.1-Venus(155-239)-G$\beta_1$ (plasmid) | *Hollins et al., 2009* | | For the mammalian expression of G$\beta_1$ tagged with a fragment of Venus (VC-G$\beta_1$). Provided by N. Lambert (Augusta University, Augusta, GA) |
| Recombinant DNA reagent | pcDNA3.1-Venus(1-155)-G$\gamma_2$ (plasmid) | *Hollins et al., 2009* | | For the mammalian expression of G$\gamma_2$ tagged with a fragment of Venus (VN-G$\gamma_2$). Provided by N. Lambert (Augusta University, Augusta, GA) |

*Continued*

| Reagent type (species) or resource | Designation | Source or reference | Identifiers | Additional information |
|---|---|---|---|---|
| Recombinant DNA reagent | pcDNA3.1-Gβ$_1$ (plasmid) | *Hollins et al., 2009* | | For the mammalian expression of untagged Gβ$_1$. Provided by N. Lambert (Augusta University, Augusta, GA) |
| Recombinant DNA reagent | pcDNA3.1-Gγ$_2$ (plasmid) | *Hollins et al., 2009* | | For the mammalian expression of untagged Gγ$_2$. Provided by N. Lambert (Augusta University, Augusta, GA) |
| Recombinant DNA reagent | pcDNA3-Gαi3 (plasmid) | *Garcia-Marcos et al., 2010* | | For the mammalian expression of rat Gαi3 |
| Recombinant DNA reagent | pcDNA3.1(-)-Gαi3-YFP | *Marivin et al., 2016* | | Citrine variant of YFP inserted in the αb/αc loop of Gαi3 |
| Recombinant DNA reagent | pcDNA3.1-masGRK3ct-Nluc (plasmid) | *Masuho et al., 2015* | | Provided by K. Martemyanov (Scripps Research Institute, Jupiter, FL) |
| Recombinant DNA reagent | pcDNA3.1-mas-KB-1753-Nluc (plasmid) | *Maziarz et al., 2020* | | |
| Recombinant DNA reagent | pcDNA3.1-Nluc-EPAC-VV (plasmid) | *Masuho et al., 2015* | | Provided by K. Martemyanov (Scripps Research Institute, Jupiter, FL) |
| Recombinant DNA reagent | pCS2+−6xMyc-GRK2ct-PM | This paper | | Contains bovine GRK2 aa495-689 fused to human Rit aa185-247 See details in 'Plasmids' section of 'Materials and methods' |
| Recombinant DNA reagent | pcDNA3.1-3xHA-M4R (plasmid) | cDNA Resource Center at Bloomsburg University | Cat# MAR040TN00 | |
| Recombinant DNA reagent | pcDNA6A-EGFR (plasmid) | Addgene | Cat# 42665 | |
| Recombinant DNA reagent | Grb2-GBA | *Parag-Sharma et al., 2016* | | Contains Grb2 fused to GIV aa1660-1705 |
| Chemical compound, drug | NanoGlo Luciferase Assay System | Promega | Cat# N1120 | |
| Chemical compound, drug | Carbachol | Acros Organics | Cat# AC-10824 | |

*Continued on next page*

Continued

| Reagent type (species) or resource | Designation | Source or reference | Identifiers | Additional information |
|---|---|---|---|---|
| Chemical compound, drug | Ramapycin | Alfa Aesar | Cat# J62473 | |
| Chemical compound, drug | Forskolin | Trocis | Cat# 1099 | |
| Chemical compound, drug | Pertussis Toxin | List Biologicals | Cat#179A | |
| Chemical compound, drug | EGF | Gold Biotechnology | Cat# 1150-04-100 | |

## Reagents

Unless otherwise indicated, all chemical reagents were obtained from Sigma-Aldrich or Fisher Scientific. Rapamycin was purchased from Alfa Aesar (#J62473) and carbachol from Acros Organics (#AC-10824). PTX was obtained from List Biologicals (#179A) and forskolin from Tocris Bioscience (#1099). Human EGF was from Gold Biotechnology (#1150-04-100) and α-hemolysin from Sigma-Aldrich (#H9395).

## Plasmids

The plasmids encoding FKBP-fused constructs were generated by replacing the pseudojanin sequence between the NruI/BamHI sites of pmRFP-FKBP-pseudojanin (Addgene, #37999) by different sequences: for pmRFP-FKBP-GIV GBA it was human GIV amino acids 1660–1705 (*Parag-Sharma et al., 2016*); for pmRFP-FKBP-AGS1* it was full length rat AGS1 (aka DEXRAS) bearing a C278S mutation to disrupt its CAAX box that allows membrane targeting; for pmRFP-FKBP-Ric-8A* it was rat Ric-8A amino acids 12–492 (*Maziarz et al., 2020*; *Thomas et al., 2011*); for pmRFP-FKBP-R12 GL it was mouse RGS12 amino acids 1185–1221 (*Maziarz et al., 2020*); and for pmRFP-FKBP-GIV-CT it was human GIV amino acids 1660–1870 (*Parag-Sharma et al., 2016*). In all cases, the sequence of the G-protein regulator was separated from the FKBP domain by the flexible linker sequence SAGGSAGGSAGGSAGGSAGGPRAQASRGSG. The plasmid encoding Lyn11-FRB has been described previously (*Parag-Sharma et al., 2016*). The plasmid encoding bovine GRK3ct (aa 495–688) fused to nanoluciferase and a membrane anchoring sequence (mas) (pcDNA3.1-masGRK3ct-NanoLuc) used as the BRET donor component in the Gβγ biosensor was a kind gift from K. Martemyanov (Scripps Research Institute, Jupiter, FL) (*Masuho et al., 2015*; *Posokhova et al., 2013*), and plasmids encoding the BRET acceptor Venus-tagged Gβγ (pcDNA3.1-Venus[1-155]-Gγ2[VN-Gγ2] and pcDNA3.1-Venus[155-239]-Gβ1[VC-Gβ1]) were kindly provided by N. Lambert (Augusta University, Augusta, GA) (*Hollins et al., 2009*; *Qin et al., 2011*). The generation of the plasmid encoding KB-1753-Nluc with a membrane anchoring sequence (pcDNA3.1-mas-KB-1753-Nluc) used as the BRET donor in the Gαi-GTP biosensor has been described previously (*Maziarz et al., 2020*), and the plasmid expressing the Gαi3 construct internally tagged with YFP at the 'b/c loop' (Gαi3-YFP) used as the BRET acceptor has also been described elsewhere (*Marivin et al., 2016*). The plasmid encoding untagged rat Gαi3 (pcDNA3-Gαi3) has been previously described (*Garcia-Marcos et al., 2011b*; *Ghosh et al., 2008*) and the plasmids encoding untagged human Gβ1 (pcDNA3.1-Gβ1), and untagged human Gγ2 (pcDNA3.1-Gγ2) were kindly provided by N. Lambert (Augusta University, Augusta, GA) (*Hollins et al., 2009*; *Qin et al., 2011*). The plasmid encoding the human M4R was obtained from the cDNA Resource Center at Bloomsburg University (pcDNA3.1-3xHA-M4R, cat# MAR040TN00). The plasmid encoding the cAMP biosensor Nluc-EPAC-VV (pcDNA3.1-Nluc-EPAC-VV) (*Masuho et al., 2015*) was a gift from K. Martemyanov (Scripps Research Institute, Jupiter, FL). The plasmid encoding EGFR (pcDNA6A-EGFR) was obtained from Addgene (#42665). The plasmids encoding Grb2-GBA WT and Grb2-GBA FA have been described previously (*Parag-Sharma et al., 2016*). The plasmid encoding GRK2ct was generated by inserting a sequence of bovine GRK2 amino acids 495–689 (GRK2ct) fused to a plasma membrane targeting sequence (human Rit amino acids 185–247) provided by P. Wedegaertner (Thomas Jefferson

University, Philadelphia, PA) (*Irannejad and Wedegaertner, 2010*) into SdaI/SmaI sites of a pCS2 + plasmid that places a 6xMyc tag sequence at the N-terminus of the insert (*Marivin et al., 2019*).

## G-protein live-cell BRET measurements

HEK293T cells (ATCC, cat# CRL-3216) were grown at 37°C, 5%CO$_2$ in high-glucose Dulbecco's modified eagle medium (DMEM) supplemented with 10% FBS, 100 U/ml penicillin, 100 µg/ml streptomycin, and 1% L-glutamine. HEK293T cells were not authenticated by STR profiling or tested for mycoplasma contamination. Approximately 400,000 cells/well were seeded on 6-well plates coated with 0.1% gelatin and transfected ~24 hr later using the calcium phosphate method. For experiments aimed at detecting free Gβγ, cells were transfected with the following amounts of plasmid DNA per well: 1 µg for Gαi3, 0.2 µg for VC-Gβ$_1$, 0.2 µg VN-Gγ$_2$, and 0.1 µg of mas-GRK3ct-Nluc. For experiments aimed at detecting Gαi-GTP, cells were transfected with the following amounts of plasmid DNA per well: 1 µg for Gαi3-YFP, 0.2 µg for Gβ$_1$, 0.2 µg Gγ$_2$, and 0.1 µg of mas-KB-1753-Nluc. For either Gβγ or Gαi-GTP measurements, cells were co-transfected with the following amounts of plasmid DNA per well: 3 µg for Lyn11-FRB, 0.2 µg for M4R, 0.5 µg for FKBP-GIV GBA, 0.125 µg for FKBP-AGS1*, 0.05 µg for FKBP-Ric8A*, 0.1 µg for FKBP-R12 GL, and 0.5 µg for FKBP-GIV-CT. Total DNA amount per well was equalized by supplementing with empty pcDNA3.1 as needed. For experiments shown in *Figure 3—figure supplement 1*, Lyn11-FRB and FKBP-fused constructs were omitted and the following amounts of plasmid DNA transfected instead: 1 µg for EGFR and 2 µg for Grb2-GBA.

Approximately 18–24 hr after transfection, cells were washed with PBS, harvested by gentle scraping, and centrifuged for 5 min at 550 × *g*. Cells were resuspended in assay buffer (140 mM NaCl, 5 mM KCl, 1 mM MgCl$_2$, 1 mM CaCl$_2$, 0.37 mM NaH$_2$PO$_4$, 20 mM HEPES pH 7.4, 0.1% glucose) at a concentration of approximately 1 million cells/ml; 25,000–50,000 cells were added to a white opaque 96-well plate (Opti-Plate, Perkin Elmer) and mixed with the nanoluciferase substrate Nano-Glo (Promega, cat# N1120, final dilution 1:200) for 2 min before measuring luminescence signals in a POLARstar OMEGA plate reader (BMG Labtech) at 28°C. Luminescence was measured at 460 ± 40 and 535 ± 10 nm, and BRET was calculated as the ratio between the emission intensity at 535 ± 10 nm divided by the emission intensity at 460 ± 40 nm. For kinetic BRET measurements, luminescence signals were measured every 0.24 s for the duration of the experiment. Reagents were added to the wells during live measurements using injectors. Kinetic measurement data are presented as the BRET change relative to the baseline signal (the average BRET ratio of the 30 s prestimulation). For endpoint measurements shown in *Figure 1—figure supplement 1*, data is presented as raw BRET ratios (535 nm luminescence/460 nm luminescence) of unstimulated cells. For *Figure 2—figure supplement 1*, PTX treatments consisted of overnight incubations with 0.2 µg/ml of the toxin. For *Figure 4*, cells were nucleotide-depleted by following a previously described protocol (*Qin et al., 2011*; *Qin et al., 2008*). Procedures were as described above except that cells were resuspended in a different assay buffer (140 mM potassium gluconate, 5 mM KCl, 10 mM HEPES, 1 mM EGTA, 0.3 mM CaCl$_2$, 1 mM MgCl$_2$, pH 7.2), and treated with 1000 U/ml of α-hemolysin and 5 mM KCN for 10 min before the start of the measurements to semi-permeabilize cells and block nucleotide synthesis, respectively. As indicated in the figures, 0.25 mM GTP was added in some cases for 2 min before stimulation with rapamycin or carbachol.

At the end of some BRET experiments, a separate aliquot of the same pool of cells used for the luminescence measurements was centrifuged for 1 min at 14,000 × *g* and pellets stored at −20°C for subsequent immunoblot analysis (see 'Protein electrophoresis and Immunoblotting' section below).

## cAMP live-cell BRET measurements

HEK293T cells were seeded and transfected using the calcium phosphate method as in 'G-protein live-cell BRET measurements' section but using the following amounts of plasmid DNA per well: 3 µg for Lyn11-FRB, 0.5 µg for FKBP-GIV GBA, 0.05 µg for Nluc-EPAC-VV, and 2 µg for GRK2ct. Total DNA amount per well was equalized by supplementing with empty pcDNA3.1 as needed. Luminescence measurements were also carried out as described in 'G-protein live-cell BRET measurements' section, except that signals were measured every 4 s instead of every 0.24 s. Results were presented as the inverse of the BRET ratio after subtraction of the basal BRET signal measured

for 60 s before any stimulation (BRET change$^{-1}$). Inhibition of forskolin-induced cAMP after 15 min was determined by calculating the difference in BRET changes with and without rapamycin addition after subtraction of a baseline signal measured in parallel with unstimulated cells. Samples for immunoblotting were prepared as in 'G-protein live-cell BRET measurements' and processed as described in 'Protein electrophoresis and Immunoblotting' section below.

## Protein electrophoresis and immunoblotting

Pellets of HEK293T cells used in BRET experiments were resuspended on ice with lysis buffer (20 mM Hepes, pH 7.2, 5 mM Mg(CH$_3$COO)$_2$, 125 mM K(CH$_3$COO), 0.4% (v:v) TritonX-100, 1 mM DTT, 10 mM β-glycerophosphate, and 0.5 mM Na$_3$VO$_4$ supplemented with a protease inhibitor cocktail [SigmaFAST, cat# S8830]). Lysates were cleared by centrifugation (10 min at 14,000 $\times$ g, 4°C) and boiled for 5 min in Laemmli sample buffer before protein separation by SDS-PAGE and electrophoretic transfer to PVDF membranes for 2 hr. PVDF membranes were blocked with TBS supplemented with 5% non-fat dry milk for 1 hr, and then incubated sequentially with primary and secondary antibodies. Primary antibody species, vendors, and dilutions were as follows: RFP (Rabbit, Rockland Immunochemicals, #600-401-379), 1:1000; Gαi3 (Rabbit, Santa Cruz Biotechnology, #sc-262), 1:250; pan-Gβ (Rabbit, Santa Cruz Biotechnology, #sc-261), 1:250; α-tubulin (Mouse, Sigma-Aldrich, #T6074); HA, 1:1000; GFP (Mouse, Clontech/Takara Bio, #632380), 1:1000; Myc (Mouse, Cell Signaling Technologies, #2276), 1:1000. Secondary antibodies (goat anti-rabbit conjugated to AlexaFluor 680 [Life Technologies, #A-21077] or goat anti-mouse conjugated to IRDye 800 [LI-COR Biosciences, #926–32210]) were used at 1:10,000. Infrared imaging of immunoblots was performed according to manufacturer's recommendations using an Odyssey CLx infrared imaging system (LI-COR Biosciences). Images were processed using the ImageJ software (NIH) and assembled for presentation using Photoshop and Illustrator software (Adobe).

## Acknowledgements

This work was supported by NIH grants R01GM136132 and R01NS117101. I thank Kirill Martemyanov (Scripps Research Institute), Philip Wedegaertner (Thomas Jefferson University), and Nevin Lambert (Augusta University) for providing plasmids. I also thank Marcin Maziarz for help with cell culture and for comments on the manuscript, and Maria Papakonstantinou for help with cell culture.

## Additional information

### Funding

| Funder | Grant reference number | Author |
| --- | --- | --- |
| National Institute of General Medical Sciences | R01GM136132 | Mikel Garcia-Marcos |
| National Institute of Neurological Disorders and Stroke | R01NS117101 | Mikel Garcia-Marcos |

The funders had no role in study design, data collection and interpretation, or the decision to submit the work for publication.

### Author contributions

Mikel Garcia-Marcos, Conceptualization, Formal analysis, Funding acquisition, Investigation, Methodology, Writing - original draft, Project administration

### Author ORCIDs

Mikel Garcia-Marcos ⓘ https://orcid.org/0000-0001-9513-4826

### Decision letter and Author response

Decision letter https://doi.org/10.7554/eLife.65620.sa1
Author response https://doi.org/10.7554/eLife.65620.sa2

## Additional files

### Supplementary files

• Transparent reporting form

### Data availability

All data generated or analysed during this study are included in the manuscript and supporting files.

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
