## [Decision Letter]

**Decision letter after peer review:**

Thank you for submitting your article "Complementary biosensors reveal different G-protein signaling modes triggered by GPCRs and non-receptor activators" for consideration by *eLife*. Your article has been reviewed by 3 peer reviewers, including Volker Dötsch as the Reviewing Editor and Reviewer #1, and the evaluation has been overseen by Jonathan Cooper as the Senior Editor. The following individual involved in review of your submission has agreed to reveal their identity: Sudarshan Rajagopal (Reviewer #2).

One issue that was discussed in particular between the reviewers is the relevance of the proposed method to normal function in light of the overexpression used and the truncation of full-length proteins.

Essential revisions:

1) It would be good to somehow get a handle at how "out of range" the expression levels for these proteins are. How much more GIV is being produced compared to native? How much native GIV is membrane targeted as opposed to elsewhere? Has a CRISPR knock-in of the FA mutant of GIV been tested?

2) Although the BRET reporter system has been described before, it would be good still to describe it in Figure 1, so we don't have to dig in the experimentals to get a sense of how the various players are labeled.

3) The complex formation should not be described as "irreversible": "In one, the GDI R12 GL reduces the availability of Gα(GDP)-Gβγ by irreversibly binding to Gα-GDP." Unless it is covalent, it is reversible. It could be effectively irreversible if the affinities and protein concentrations are high enough (unclear that is the case here although reported affinities are order of magnitude better than that of GIV).

4) In Figure 6C the GIV data need some clarification. It appears because carbachol induces a similar shift in BRET in the presence of GIV compared to the GPCR alone, it is said that GIV does not hinder activation of the GPCR. It would seem that the pool of Gbg being released by GIV (before addition of carbachol) would be different than the pool of Gbg that the receptor liberates? This is different from the other three cases where the total BRET signal ends up being the same regardless of the protein. Why the difference?

5) It would be better to report SD rather than SE. SD measures the amount of variability, or dispersion, from the individual data values to the mean, while the standard error measures how far the sample mean (average) of the data is likely to be from the true population mean. In these experiments, one is more interested in the SD.

6) Gbg is also required for recruitment of GRK2. Has the author assessed GRK2 activity in this setting?

Reviewer #1:

Mikel Garcia-Marcos describes in this manuscript two different aspects: First he introduces a new method that can be used to investigate the effect of cellular effector proteins on the activation of G-proteins. This method is based on induced hetero-dimerization using the small drug rapamycin that has been established many years ago. He uses this system to recruit proteins with a presumed GEF activity to the membrane where they can interact with heterotrimeric G-proteins. The effect he measures on identifying the concentration of Gbeta/γ and Galpha-GTP.

In the second part he uses this system to investigate the effect of three GEF proteins on membrane-anchored Gi protein. He finds that within the group of three GEF proteins (GIV, AGS1 and Ric-8a), GIV promotes activation by dissociation of the Gbeta/γ dimer but not by formation of Galpha-GTP despite its in vitro GEF function. This result is surprising but the data are compelling.

Overall, the method is interesting, enlarging the tool box for investigating the activation mechanism of G-proteins. The data on the different GEF proteins are likewise interesting and within the framework of this assay plausible.

Reviewer #2:

This is an interesting manuscript that addresses a very important question in the field of G-protein signaling – whether their activation by G protein-coupled receptors (GPCRs) is similar to other activators of G protein signaling such as GIV, AGS1, etc. Such an analysis has previously been limited by a lack of tools to detect free Gbg and Ga-GTP formation. The author uses novel biosensors of Gai-GTP and Gbg to probe this system. The author finds that GIV, unlike other GBA proteins, activates G protein signaling in cells primarily through the formation of free Gbg rather than through the formation of Gai-GTP, although it has GEF activity in vitro. This is unlike AGS1, which triggers the formation of Gai-GTP. Notably, both R12 and AGS1 hinder activation of G proteins while GIV does not. This clearly demonstrates that activation of heterotrimeric G proteins can occur through multiple mechanisms with different signaling outcomes. Notably, a larger role for Gbg is appreciated in promoting signaling through a variety of pathways, including inhibition of adenylyl cyclase.

Reviewer #3:

In this paper, the author sought to study the ability of a series of non-receptor GEFs, in particular GIV/girdin, to activate both Galpha subunits and Gbg subunits under more physiological settings. About 11 years ago the lead author reported GEF activity by GIV using purified components, although this activity was lower than that mediated by GPCRs and at orders of magnitude higher EC50. Because GIV and some other non-receptor regulators (i.e. AGS1, and RGS12) can possess the ability to displace Gbg subunits, a key question that could be addressed with these experiments is whether it is the released Gbg subunits or the GEF activity that is important for GIV function in cells.

The strength of the approach used here is that the author can trigger recruitment of the G protein binding domains of GIV and other proteins by addition of rapamycin, which allows one to study that binding interaction in the absence of the many other interactions that could be formed by these proteins in cells. The setting is more physiological than when using purified components but key weaknesses remain in that (a) all the proteins in the system are being over expressed relative to native levels, and (b) it requires truncations that have fewer competing interactions that could arguably prevent the proteins from interacting at all if they were present at native concentrations. The bottom line is that the experiments are still far from physiological.

That said, the BRET assay data look clean, reasonable control experiments are run, and they together give a surprising result in that, within the context of these experiments, there is little or no GEF activity provided by GIV, but that it can release Gbg. This is a bit of a paradigm shift for the GIV non-receptor GEF field , which has lately been the domain of two alumni from the Fahrquar lab where studies of the protein originated. In many papers to date, the underlying hypothesis from these labs, even in 2020, has been that the GEF activity of GIV drives its physiological effects. Thus it is quite admirable to perform and publish a more definitive experiment even when it goes against the standard mantra. That's good science. The data in Figure 3 was particularly illuminating, where inhibition of cAMP production was eliminated by GRK2ct, showing in this context that it is not a result of GIV GEF activity on Gi.

The methods deployed in this study should be useful to the field, in particular the rapamycin-based recruitment which could add a new dimension to cell based studies where one normally just overexpresses the proteins of interest. Its impact however is questionable because one is now potentially targeting the domains to regions they would not necessarily go because the proteins are not full length, especially when they have many other reported interaction partners and furthermore when they are being over expressed so one can measure significant BRET signal in the first place.

Impact is somewhat further diminished by some backpedalling on the GEF story based on data in Figures 5 and 6. The author concludes that there may still be meaningful GEF activity mediated by GIV in vitro that serves to prevent GIV from holding on to Galpha subunits too long (they do not bind Ga-GTP). This is an attempt to explain why GIV has no effect on GPCR signaling function, while other proteins tested (AGS1, R12) do. A more parsimonious conclusion would be that GIV just doesn't have meaningful GEF activity in cells, even when overexpressed as a membrane-recruitable fragment. This seems particularly likely considering that GIV has lower affinity for Galpha subunits relative to GPCRs by orders of magnitude, and with a high nM EC50 that seems unlikely to function at physiological concentrations of these proteins.

[Editors' note: further revisions were suggested prior to acceptance, as described below.]

Thank you for resubmitting your work entitled "Complementary biosensors reveal different G-protein signaling modes triggered by GPCRs and non-receptor activators" for further consideration by *eLife*. Your revised article has been evaluated by Jonathan Cooper (Senior Editor) and a Reviewing Editor.

The manuscript has been improved but there are some remaining issues that need to be addressed, as outlined below:

Reviewer #2:

The revisions have addressed my concerns.

Reviewer #3:

Dr. Garcia-Marcos has addressed many of the concerns brought up by my prior review. The paper is now easier to interpret and the figures are first class.

I have mixed feelings about impact. I like the fact that the work represents a paradigm shift, straight from the lab of the world leader on this topic. The observation that this fragment of GIV can efficiently release Gbg subunits could be a game changer. On the other hand, this is a bit of a niche field in the realm of heterotrimeric G protein signaling, and many are agnostic about the ability of GIV to serve as a GEF in the first place. This is in part because a lot of the evidence comes from studies that use fragments or overexpression etc. The killer in vivo experiment has not yet been performed (a GIV knock-in that eliminates the proposed function of the GIV domain in question at physiological levels). And if one is agnostic about this issue, and if one thinks of it as more "niche", then it is not much of a paradigm shift.

A real strength from a technical perspective here is the parallel examination of different soluble regulators of heterotrimeric G proteins using the same rapamycin membrane recruitment system. The author has done a good job describing the caveats in interpreting the data because it is true that the system, especially for GIV, is a long way from physiological…GIV has many proposed interaction partners in the cell. However, the results are interesting because of the dramatically different effects on observes in the system with the various proteins and the potential for their use, at the very least, and chemical biological tools to probe G protein signaling. I agree with the author that the best way to sell this work is as an effort to characterize possible mechanisms in a cellular context. One might argue this is still "in vitro", but the fact that quite different answers are achieved here versus the test tube is interesting.

Only one major remaining concern. I understand the author's desire in trying to incorporate the GEF activity that has been observed with purified fragments at high concentrations in the test tube or at low levels in living cells into a working model. However, upon examining all the data in this paper, there seems to be little or no evidence here that this fragment of GIV is leading to any nucleotide exchange on Gi in the cell. And this is even after overexpression of a fragment free from other interactions in the cell. The rapamycin treatments with the Gi sensor are completely inert when it comes to GIV. Unless one argues that the Gi sensor is just not that great and the signal gets lost in the noise (and I am not sure it is wise to make that argument). Regardless, the data in this paper just doesn't support a model where GEF activity factors in (Figure 6). The GIV fragment just seems really good at liberating Gbg by some other mechanism than GTP loading, at least when it is membrane targeted in this way.

The thoughtful discussion does come up with a few ideas about how weak GEF activity might still play into the GIV system; it is just that to me the paper seems to make such speculation unnecessary. I would be comfortable with a much more simple conclusion that membrane recruitment of GIV by any mechanism could lead to Gbg release in the absence of GPCRs. That's interesting and simple.

It could be that the previously observed low "GEF activity" in living cells is a consequence of release of the tonic GDI activity of Gbg, allowing Gi to exchange on its own. But I'd have to dig into the papers to figure out what the controls were. I am just speculating.

Typos. line 375: nor should be "not". Line 370: "in controlled" should be "is controlled".

---

## [Author Response]

Essential revisions:1) It would be good to somehow get a handle at how "out of range" the expression levels for these proteins are. How much more GIV is being produced compared to native? How much native GIV is membrane targeted as opposed to elsewhere? Has a CRISPR knock-in of the FA mutant of GIV been tested?

This point has multiple parts. I will respond to them separately.

How much more GIV is being produced compared to native?

This has not been possible to assess because we lack an antibody that can detect the fragment of GIV that is used in the ectopically expressed construct.

However, I would like to clarify that the purpose of the experimental system used here was not to closely mimic the properties of the native protein, but to assess more thoroughly the mechanism of its G protein regulatory activity in a cellular context. The role of native GIV in G protein-mediated signaling has been extensively studied by my laboratory and others (partially reviewed in (Aznar et al., 2016; Garcia-Marcos et al., 2015)), but the key question that had been elusive until now was about the precise mechanism by which it activates G protein signaling in cells. I agree that the system implemented here has limitations, including the expression of ectopic GPCRs, G proteins, and fragments of GIV, but it allows the precise dissection of the mechanism of G protein activation in cells that is not attainable through other approaches. Thus, to account for the limitations of the system, the approach was to compare GIV with GPCRs and other G protein regulators side by side under the same experimental conditions. It should be noted that the motif of GIV used in our construct has been previously shown to be necessary and sufficient to regulate G proteins in vitro, same as with the fragments of RGS12 and Ric-8A or with non-prenylated AGS1. As discussed below, there is also strong evidence that the GBA motif of GIV is important for regulating G protein signaling under native conditions.

The manuscript has been modified to acknowledge more clearly these limitations. In addition to the already existing section in Discussion (end of first paragraph), I have included a new section in Results (the paragraph that starts on Page 4 and finishes of Page 5).

How much native GIV is membrane targeted as opposed to elsewhere?

This has been investigated in previous work (Parag-Sharma et al., 2016) and the conclusion is that GIV is largely excluded from membranes under normal culture conditions. Briefly, three different cell lines were analyzed for the distribution of GIV between cytosol (100,000 xg supernatant, S100) and particulate (100,000 xg pellet, P100) cell fractions. Depending on the cell line, between 50% and 90% of the GIV was recovered in the cytosolic fraction, and the pool in the P100 fraction was insoluble in non-ionic detergent, suggesting that it associates with the actin cytoskeleton rather than with membranes. These results can be found in Figure 1 of the paper in the links below, where additional details on methods and interpretations are provided (Parag-Sharma et al., 2016).

https://www.sciencedirect.com/science/article/pii/S0021925820343830 https://pubmed.ncbi.nlm.nih.gov/27864364/

Previous work has also shown that GIV can re-localize from the cytosol to the plasma membrane or to receptors localized at the plasma membrane upon ligand mediated stimulation. A clear example is shown in Figure 5 of the paper in the links below (Leyme et al., 2015).

https://rupress.org/jcb/article/210/7/1165/38482/ https://pubmed.ncbi.nlm.nih.gov/26391662/

This point has now been clarified in the text (Discussion, end of the first paragraph on Page 11).

Has a CRISPR knock-in of the FA mutant of GIV been tested?

No, a CRISPR knock-in of the FA mutant has not been tested. However, the same idea has been thoroughly tested in the past using rigorous approaches available at the time. More specifically, the role of GIV’s GBA motif has been investigated in multiple signaling contexts using “rescue” experiments with the FA mutant in GIV-depleted cellular backgrounds. Essentially, the ability of full-length, RNAi-resistant GIV WT vs. FA mutant to restore signaling defects associated with GIV knockdown has revealed that G protein regulation by GIV is critical for several signaling responses in multiple cell lines (Garcia-Marcos et al., 2011; Garcia-Marcos et al., 2009; Garcia-Marcos et al., 2012; Leyme et al., 2016; Leyme et al., 2015; Lopez-Sanchez et al., 2014; Lopez-Sanchez et al., 2015; Ma et al., 2015; Midde et al., 2015; Sasaki et al., 2015). Similar approaches have been used with the GIV-related protein DAPLE in cells or even in whole organisms (Aznar et al., 2015; Marivin et al., 2019). Other complementary approaches have further validated the involvement of GIV’s G protein regulatory function in signaling. These include knock-down/ rescue experiments with a Gαi mutant that cannot bind GIV (while retaining other normal functions and regulation by other proteins) (Garcia-Marcos et al., 2010), or a rationally-engineered protein that specifically binds and inhibits GBA motifs (Leyme et al., 2017).

Overall, there is abundant and rigorous evidence to support the role of GIV’s G protein regulatory function in cellular signaling.

2) Although the BRET reporter system has been described before, it would be good still to describe it in Figure 1, so we don't have to dig in the experimentals to get a sense of how the various players are labeled.

The BRET reporter systems for free Gβγ and GTP-bound Gαi are now depicted in Figure 1 and described in its legend.

3) The complex formation should not be described as "irreversible": "In one, the GDI R12 GL reduces the availability of Gα(GDP)-Gβγ by irreversibly binding to Gα-GDP." Unless it is covalent, it is reversible. It could be effectively irreversible if the affinities and protein concentrations are high enough (unclear that is the case here although reported affinities are order of magnitude better than that of GIV).

I agree that this was not the right choice of words. The text has now modified to avoid using “irreversible” (see changes on Pages 8 and 9). I hope the new language is clear enough and accurate.

4) In Figure 6C the GIV data need some clarification. It appears because carbachol induces a similar shift in BRET in the presence of GIV compared to the GPCR alone, it is said that GIV does not hinder activation of the GPCR. It would seem that the pool of Gbg being released by GIV (before addition of carbachol) would be different than the pool of Gbg that the receptor liberates? This is different from the other three cases where the total BRET signal ends up being the same regardless of the protein. Why the difference?

I am not sure if I understand the comment, but I will try to clarify the interpretation of results. I believe that the differences emerge from how the different regulators affect the availability of Gαβγ trimers for GPCR-mediated activation. We cannot distinguish “pools” of Gβγ that are being released because this is a system in dynamic equilibrium, i.e., Gβγ (and Gαi-GTP) levels observed are a result of multiple turnover cycles of Gαβγ by GPCRs and/or other regulators. If GPCR-induced changes in Gβγ (or Gαi-GTP) are reduced after the action of a regulator, we interpret that the pool of Gαβγ available for activation under the new dynamic equilibrium condition promoted by a regulator (R12 GL, GIV, AGS1) has been reduced. This occurs for R12 GL and AGS1 but not for GIV. R12 GL can lock Gα-GDP in a Gβγ-dissociated state, and AGS1 can deplete Gαβγ by using it as substrate to catalyze multiple turnover cycles of nucleotide exchange, whereas GIV does not do either based on the evidence presented in the manuscript.

The reviewers might be specifically thinking about the observation that final BRET levels post-GPCR stimulation are higher in the GIV condition than in the other ones. This implies that the diminished response after GPCR stimulation with R12 GL or AGS1 is not due to reaching a limit of how much Gβγ can be released and/or detected under these experimental conditions.

As requested, the section related to Figure 6 has been re-written to clarify these points (see changes in the last section of Results, on Pages 8 and 9).

5) It would be better to report SD rather than SE. SD measures the amount of variability, or dispersion, from the individual data values to the mean, while the standard error measures how far the sample mean (average) of the data is likely to be from the true population mean. In these experiments, one is more interested in the SD.

All figures have now been modified to report SD instead of SE.

6) Gbg is also required for recruitment of GRK2. Has the author assessed GRK2 activity in this setting?

I have not directly assessed GRK2 activity in this setting, but the free Gβγ BRET biosensor used throughout the manuscript is based on measuring the interaction between Gβγ and GRK3, which closely resembles the interaction between Gβγ and GRK2. However, I would like to indicate that GRK2/3 activity is not only regulated by Gβγ, but also by interactions with GPCRs. As exemplified by observations in a recent *eLife* paper https://elifesciences.org/articles/54208 (Stoeber et al., 2020), the relative contribution of Gβγ vs. GPCR interactions to GRK2 activation is still poorly understood and beyond the scope of the current manuscript, which is focused on the direct effect of various regulators on G proteins rather than downstream events.

Reviewer #1:Mikel Garcia-Marcos describes in this manuscript two different aspects: First he introduces a new method that can be used to investigate the effect of cellular effector proteins on the activation of G-proteins. This method is based on induced hetero-dimerization using the small drug rapamycin that has been established many years ago. He uses this system to recruit proteins with a presumed GEF activity to the membrane where they can interact with heterotrimeric G-proteins. The effect he measures on identifying the concentration of Gbeta/γ and Galpha-GTP.In the second part he uses this system to investigate the effect of three GEF proteins on membrane-anchored Gi protein. He finds that within the group of three GEF proteins (GIV, AGS1 and Ric-8a), GIV promotes activation by dissociation of the Gbeta/γ dimer but not by formation of Galpha-GTP despite its in vitro GEF function. This result is surprising but the data are compelling.Overall, the method is interesting, enlarging the tool box for investigating the activation mechanism of G-proteins. The data on the different GEF proteins are likewise interesting and within the framework of this assay plausible.

I appreciate the overall positive tone of these comments.

Reviewer #2:This is an interesting manuscript that addresses a very important question in the field of G-protein signaling – whether their activation by G protein-coupled receptors (GPCRs) is similar to other activators of G protein signaling such as GIV, AGS1, etc. Such an analysis has previously been limited by a lack of tools to detect free Gbg and Ga-GTP formation. The author uses novel biosensors of Gai-GTP and Gbg to probe this system. The author finds that GIV, unlike other GBA proteins, activates G protein signaling in cells primarily through the formation of free Gbg rather than through the formation of Gai-GTP, although it has GEF activity in vitro. This is unlike AGS1, which triggers the formation of Gai-GTP. Notably, both R12 and AGS1 hinder activation of G proteins while GIV does not. This clearly demonstrates that activation of heterotrimeric G proteins can occur through multiple mechanisms with different signaling outcomes. Notably, a larger role for Gbg is appreciated in promoting signaling through a variety of pathways, including inhibition of adenylyl cyclase.

I appreciate the overall positive tone of these comments.

Reviewer #3:In this paper, the author sought to study the ability of a series of non-receptor GEFs, in particular GIV/girdin, to activate both Galpha subunits and Gbg subunits under more physiological settings. About 11 years ago the lead author reported GEF activity by GIV using purified components, although this activity was lower than that mediated by GPCRs and at orders of magnitude higher EC50. Because GIV and some other non-receptor regulators (i.e. AGS1, and RGS12) can possess the ability to displace Gbg subunits, a key question that could be addressed with these experiments is whether it is the released Gbg subunits or the GEF activity that is important for GIV function in cells.The strength of the approach used here is that the author can trigger recruitment of the G protein binding domains of GIV and other proteins by addition of rapamycin, which allows one to study that binding interaction in the absence of the many other interactions that could be formed by these proteins in cells. The setting is more physiological than when using purified components but key weaknesses remain in that (a) all the proteins in the system are being over expressed relative to native levels, and (b) it requires truncations that have fewer competing interactions that could arguably prevent the proteins from interacting at all if they were present at native concentrations. The bottom line is that the experiments are still far from physiological.

I agree that the conditions are still not physiological. However, these experiments provide mechanistic information that is not attainable through other approaches. I believe that the approach presented here bridges biochemical experiments with purified proteins in vitro and genetic manipulations of natively expressed proteins in cells or in vivo (which are summarized in the response to Essential revisions 1 above), providing complementary information to elucidate more precisely molecular mechanisms of G protein regulation.

That said, the BRET assay data look clean, reasonable control experiments are run, and they together give a surprising result in that, within the context of these experiments, there is little or no GEF activity provided by GIV, but that it can release Gbg. This is a bit of a paradigm shift for the GIV non-receptor GEF field , which has lately been the domain of two alumni from the Fahrquar lab where studies of the protein originated. In many papers to date, the underlying hypothesis from these labs, even in 2020, has been that the GEF activity of GIV drives its physiological effects. Thus it is quite admirable to perform and publish a more definitive experiment even when it goes against the standard mantra. That's good science. The data in Figure 3 was particularly illuminating, where inhibition of cAMP production was eliminated by GRK2ct, showing in this context that it is not a result of GIV GEF activity on Gi.

In all honesty, this reviewer’s whole assessment is one of the best constructive criticisms I have received to date. These comments are very well taken. From this section, I particularly appreciate that the reviewer finds “quite admirable to perform and publish a more definitive experiment even when it goes against the standard mantra. That's good science”. Chasing this “more definitive experiment” has been a personal goal for over a decade, so it is rewarding to read that the reviewer finds the effort admirable. In the past, the reasonable assumption made was that the effects of GIV on G protein signaling in cells could be ascribed to its GEF function because they were disrupted by a point mutation that disrupts its ability to enhance nucleotide exchange in vitro. Although we were aware of the potential caveat that the same mutation also prevents GIV-mediated release of Gβγ from Gαβγ heterotrimers, no approach was available to disentangle the mechanism. This motivated a long-term effort of my laboratory in generating live-cell biosensors for Gα-GTP. Once these new tools became available, I have been able to re-evaluate the mechanism of G protein regulation by GIV in cells.

I would like to say that Figure 3 on its own just indicates that the inhibition of cAMP mediated by GIV is Gβγ-dependent. However, regulation of different adenylyl cyclase isoforms in cells is very complex and can involve positive and negative cooperation between Gβγ and other regulatory inputs (Gα, Ca^2+^, phosphorylation). Only when interpreting these data along with the results obtained with G protein biosensors it becomes more convincing that GIV does not elevate Gαi-GTP levels under these conditions.

The methods deployed in this study should be useful to the field, in particular the rapamycin-based recruitment which could add a new dimension to cell based studies where one normally just overexpresses the proteins of interest. Its impact however is questionable because one is now potentially targeting the domains to regions they would not necessarily go because the proteins are not full length, especially when they have many other reported interaction partners and furthermore when they are being over expressed so one can measure significant BRET signal in the first place.

I cannot predict what the impact will be, but I agree that it could add a new dimension to cell based studies by permitting the interrogation of some mechanistic questions that are currently not addressable through other approaches. Thus, although I agree that the approach has limitations, as indicated in the manuscript and in the response to reviewers above, I believe that it complements other approaches typically used in the field.

Impact is somewhat further diminished by some backpedalling on the GEF story based on data in Figures 5 and 6. The author concludes that there may still be meaningful GEF activity mediated by GIV in vitro that serves to prevent GIV from holding on to Galpha subunits too long (they do not bind Ga-GTP). This is an attempt to explain why GIV has no effect on GPCR signaling function, while other proteins tested (AGS1, R12) do. A more parsimonious conclusion would be that GIV just doesn't have meaningful GEF activity in cells, even when overexpressed as a membrane-recruitable fragment. This seems particularly likely considering that GIV has lower affinity for Galpha subunits relative to GPCRs by orders of magnitude, and with a high nM EC50 that seems unlikely to function at physiological concentrations of these proteins.

I believe that the reviewer and I agree, but that there might be nuances in what “meaningful GEF activity” means for each one of us. That GIV has GEF activity when it binds to Gαi in vitro has been established in the past using “gold standard” assays (de Opakua et al., 2017; Garcia-Marcos et al., 2011; Garcia-Marcos et al., 2010; Garcia-Marcos et al., 2009; Garcia-Marcos et al., 2012; Maziarz et al., 2018). It has also been established that the binding of GIV to Gαi results in the release of Gβγ from Gαβγ trimers in vitro (Garcia-Marcos et al., 2009). Then, if GIV promotes the release of Gβγ form Gαβγ in cells as shown in this work, it must also bind to Gαi under the same conditions, regardless of the affinity of the interaction compared to Gα-GTP hat of GPCRs or other G protein regulators. It is therefore inescapable that under these conditions GIV must be exerting the GEF activity that can be measured when it binds in vitro. In my opinion, the key question is whether this GEF activity is robust enough to lead to the accumulation of Gαi-GTP in cells. The data tell us that this is not the case. If this is what the reviewer means with “meaningful GEF activity”, we agree. However, we cannot simply neglect that the data also tell us that when GIV binds to Gαi, even as a purified 1:1 equimolar complex (Garcia-Marcos et al., 2011), it enhances nucleotide exchange.

The results presented in Figure 5 and 6 just indicate a context in which this function of GIV might be relevant.

[Editors' note: further revisions were suggested prior to acceptance, as described below.]

Reviewer #3:Dr. Garcia-Marcos has addressed many of the concerns brought up by my prior review. The paper is now easier to interpret and the figures are first class.

I am glad the reviewer found that the manuscript has improved. Once again, I also appreciate the thoughtful, constructive comments. I respond below to the comments one by one. Because many of the comments and responses are more a general discussion than specific actions on the manuscript, the reviewer can scroll down response to comment #3 for a direct answer to her/his “major remaining concern”.

I have mixed feelings about impact. I like the fact that the work represents a paradigm shift, straight from the lab of the world leader on this topic. The observation that this fragment of GIV can efficiently release Gbg subunits could be a game changer. On the other hand, this is a bit of a niche field in the realm of heterotrimeric G protein signaling, and many are agnostic about the ability of GIV to serve as a GEF in the first place. This is in part because a lot of the evidence comes from studies that use fragments or overexpression etc. The killer in vivo experiment has not yet been performed (a GIV knock-in that eliminates the proposed function of the GIV domain in question at physiological levels). And if one is agnostic about this issue, and if one thinks of it as more "niche", then it is not much of a paradigm shift.

I respect the mixed feelings of the reviewer and I think I understand her/his reasons, but I would like to give my humble opinions on some of the points raised. Just to make sure that the tone of my writing is not misinterpreted, the reviewer’s comments are well taken and there is no bitterness in my discussion below.

“…many are agnostic about the ability of GIV to serve as a GEF in the first place”, “And if one is agnostic about this issue”

I have two thoughts about this. One is that the current manuscript should be of particular interest to those “many” who are agnostic, because it directly addresses a point on which they seem to have a strong opinion formed. The second thought is that the word “agnostic” resonates in my head with something that is a form of belief, and beliefs feed dogmatic unproductive attitudes in science. I think it is more productive to be a skeptic and take one step at time in regards to what can be learned from the data and approaches available at any given time in history. I hope the reviewer agrees that the work presented in this manuscript has been driven by uncompromised skepticism.

“*the work represents a paradigm shift*” and “*could be a game changer*” but “*this is a bit of a niche field*”

What is a niche field versus what is the mainstream in a field is determined to a great extent by circumstances (who, when, how, etc), so I would not take work on one or the other as an indicator of more or less potential impact. I will not dare to give specific examples, but within the field of GPCR and heterotrimeric G protein signaling there is a disproportionate number of published papers on mainstream topics that make little difference or even just muddle the waters. Thus, I will be glad with the impact of this work if it turns out to be a game changer in a given field, even if it is one considered a niche.

"The killer in vivo *experiment has not yet been performed (a GIV knock-in that eliminates the proposed function of the GIV domain in question at physiological levels).*”

I respectfully disagree with what the reviewer implies in this statement. I have two thoughts in this regard. The first thought is that we and others have performed experiments along the years with full-length GIV (and DAPLE) expressed at physiological levels to compare a mutant that disables its ability to regulate G proteins with wild-type (Aznar et al., 2015; Garcia-Marcos et al., 2011; Garcia-Marcos et al., 2009; GarciaMarcos et al., 2012; Leyme et al., 2016; Leyme et al., 2015; Lopez-Sanchez et al., 2014; Lopez-Sanchez et al., 2015; Ma et al., 2015; Marivin et al., 2019; Midde et al., 2015; Sasaki et al., 2015). The fact that genome editing is within reach nowadays does not invalidate the conclusions of previously performed experiments using other approaches as long as they are properly controlled, which I think has been the case for GIV and its G protein regulatory function.

The second thought is that performing the experiment mentioned by the reviewer (knock-in of mutation that disables the G protein regulator activity) would not only provide just a small increment over what we already know, but also that it would not address the critical question investigated in the current manuscript— i.e., whether GIV efficiently promotes nucleotide exchange in cells. In other words, we can engineer the genome of a cell or an animal to disrupt the G protein regulatory motif of GIV and observe differences in G protein signaling like the ones we have observed using other approaches, but that would not tell us is such differences in signaling are due to nucleotide exchange (Gα-GTP) or heterotrimer dissociation (free Gβγ) because the mutant does not allow to distinguish these two possibilities. This is precisely the motivation for and the advance provided by the work presented in this manuscript.

A real strength from a technical perspective here is the parallel examination of different soluble regulators of heterotrimeric G proteins using the same rapamycin membrane recruitment system. The author has done a good job describing the caveats in interpreting the data because it is true that the system, especially for GIV, is a long way from physiological…GIV has many proposed interaction partners in the cell. However, the results are interesting because of the dramatically different effects on observes in the system with the various proteins and the potential for their use, at the very least, and chemical biological tools to probe G protein signaling. I agree with the author that the best way to sell this work is as an effort to characterize possible mechanisms in a cellular context. One might argue this is still "in vitro", but the fact that quite different answers are achieved here versus the test tube is interesting.

I have added a sentence in the Discussion to clarify further the nature of the approach and its limitations. I have dubbed it a “cell-based reductionist approach”.

Only one major remaining concern. I understand the author's desire in trying to incorporate the GEF activity that has been observed with purified fragments at high concentrations in the test tube or at low levels in living cells into a working model. However, upon examining all the data in this paper, there seems to be little or no evidence here that this fragment of GIV is leading to any nucleotide exchange on Gi in the cell. And this is even after overexpression of a fragment free from other interactions in the cell. The rapamycin treatments with the Gi sensor are completely inert when it comes to GIV. Unless one argues that the Gi sensor is just not that great and the signal gets lost in the noise (and I am not sure it is wise to make that argument). Regardless, the data in this paper just doesn't support a model where GEF activity factors in (Figure 6). The GIV fragment just seems really good at liberating Gbg by some other mechanism than GTP loading, at least when it is membrane targeted in this way.The thoughtful discussion does come up with a few ideas about how weak GEF activity might still play into the GIV system; it is just that to me the paper seems to make such speculation unnecessary. I would be comfortable with a much more simple conclusion that membrane recruitment of GIV by any mechanism could lead to Gbg release in the absence of GPCRs. That's interesting and simple.

I have modified sections in the Results and the Discussion related to Figure 6 to eliminate any reference or interpretation related to possible roles of the weak GEF activity of GIV in cells. I have also simplified the conclusions according to the suggestions of this reviewer.

It could be that the previously observed low "GEF activity" in living cells is a consequence of release of the tonic GDI activity of Gbg, allowing Gi to exchange on its own. But I'd have to dig into the papers to figure out what the controls were. I am just speculating.

This is a formal possibility, but I do not think it would be possible to draw a definitive conclusion from the experiments that were performed. I would also like to say that the tonic GDI activity of Gβγ has only been shown under specific Mg^2+^ concentration conditions in vitro and that it is only partial for Gi/o α subunits (Higashijima et al., 1987; Kozasa and Gilman, 1995; Ueda et al., 1994). Even though it is widely believed that such tonic GDI activity of Gβγ exists, there is no direct evidence showing that Gβγ blocks nucleotide exchange in cells.

Typos. line 375: nor should be "not". Line 370: "in controlled" should be "is controlled".

These typos have been corrected.